# Sum-of-Squares Programming for Ma-Trudinger-Wang Regularity of Optimal Transport Maps

## Abstract

For a given ground cost, approximating the Monge optimal transport map that pushes forward a given probability measure onto another has become a staple in several modern machine learning algorithms. The fourth-order Ma-Trudinger-Wang (MTW) tensor associated with this ground cost function provides a notion of curvature in optimal transport. The non-negativity of this tensor plays a crucial role for establishing continuity for the Monge optimal transport map. It is, however, generally difficult to analytically verify this condition for any given ground cost. To expand the class of cost functions for which MTW non-negativity can be verified, we propose a provably correct computational approach which provides certificates of non-negativity for the MTW tensor using Sum-of-Squares (SOS) programming. We further show that our SOS technique can also be used to compute an inner approximation of the region where MTW non-negativity holds. We apply our proposed SOS programming method to several practical ground cost functions to approximate the regions of regularity of their corresponding optimal transport maps.

## 1 Introduction

Optimal transport (OT) (Villani, 2021; 2009; Santambrogio, 2015), originally considered by Gaspard Monge in 1781 (Monge, 1781), has become a key tool in modern machine learning with applications in generative modeling (Montavon et al., 2016; Bousquet et al., 2017; Balaji et al., 2020; Rout et al., 2021; Lipman et al., 2022; Houdard et al., 2023), adversarial training (Sanjabi et al., 2018; Bhagoji et al., 2019), control (Chen et al., 2016b; 2021; Teter et al., 2024), and data science (Peyré et al., 2019; Flamary et al., 2021). In many applications, it is of interest (Seguy et al., 2018; Makkuva et al., 2020; Bunne et al., 2022; Pooladian et al., 2023; Manole et al., 2024) to numerically approximate the Monge OT map $\tau : \mathcal{X} \rightarrow \mathcal{Y}$ where $\mathcal{X}, \mathcal{Y} \subseteq \mathcal{M}$, and $\mathcal{M}$ is a smooth, compact, connected $n$-dimensional manifold.

The Monge OT formulation is as follows: given two probability measures $\mu, \nu$ supported on $\mathcal{X}$ and $\mathcal{Y}$, respectively, let $c : \mathcal{X} \times \mathcal{Y} \rightarrow \mathbb{R}_{\geq 0}$ be a $C^4$ smooth function that encodes the cost of transporting unit amount of mass from $x \in \mathcal{X}$ to $y \in \mathcal{Y}$. The function $c$ is referred to as the *ground cost*. The problem seeks to find the Borel map $\tau_{\mathrm{opt}}$, referred to as the *Monge OT map*, which minimizes the total transportation cost, namely the functional $\int_{\mathcal{X}} c(x, \tau(x)) \mathrm{d}\mu(x)$, subject to the constraint that for all Borel sets $\mathcal{U} \subseteq \mathcal{Y}$, $\mu\left(\tau^{-1}(\mathcal{U})\right) = \nu(\mathcal{U})$, or equivalently that the pushforward $\tau_{\#}\mu = \nu$. So,

$$
\tau_{\mathrm{opt}} = \underset{\tau : \mathcal{X} \rightarrow \mathcal{Y}}{\arg\inf} \int_{\mathcal{X}} c(x, \tau(x)) \mathrm{d}\mu(x)
$$
$$
\text{subject to} \quad \tau_{\#}\mu = \nu.
$$

For a general class of costs and measures, the existence-uniqueness of the Monge OT map $\tau_{\mathrm{opt}}$ was established by (Brenier, 1991) and (Gangbo & McCann, 1996). Building on these results, it is natural to study the regularity (e.g., continuity, injectivity) of $\tau_{\mathrm{opt}}$. For the Euclidean squared-distance cost function, this was studied by (Caffarelli, 1992; Delanoë, 1991; Urbas, 1997), but for more general cost functions this remained an open problem for some time. Eventually, (Ma et al., 2005) discovered

an biquadratic form inequality–the so-called *Ma-Trudinger-Wang (MTW) condition*–which plays a crucial role in the regularity theory of $\tau_{\text{opt}}$ (Trudinger & Wang, 2009). See Sec. 2.1 for details.

However, analytic verification of this tensor non-negativity condition is tedious in practice. Consequently, existing verification results (Ma et al., 2005; Lee & McCann, 2011; Lee & Li, 2012; Figalli et al., 2012; Khan & Zhang, 2020) are tailored for specific OT problems, and the analytic approaches therein do not generalize.

**Contributions.** We propose, for the first time, a provably correct *computational* framework that can certify or falsify the non-negativity of the MTW tensor associated with a given ground cost $c$ under the assumption

**A1.** $c$ is a rational function in $(x, y) \in \mathcal{X} \times \mathcal{Y}$ semialgebraic (Definition 4).

We will see later that *necessary* condition for the proposed computational framework is that the elements of the MTW tensor are rational. For this to hold, **A1** is *sufficient but not necessary*. In fact, the development of OT regularity theory was motivated by the engineering problem of reflector antenna design (Loeper, 2009) cast as an OT problem with non-rational cost $c(x, y) = -\log \|x-y\|$, which our proposed framework can handle. Furthermore, many cost functions in practice are either already polynomials/rationals, or smooth enough to be well-approximated by polynomials/rationals. This is another reason why the assumption **A1** is benign.

The proposed approach is based on Sum-of-Squares (SOS) programming (Prajna et al., 2002; Parrilo, 2003; Prajna et al., 2005; Laurent, 2009) well-known in optimization and control literature. As such, ours is the first work on computational certification of OT regularity, and should be of interest to the broader SOS programming research community.

We also demonstrate that our proposed computational framework

- can be applied to non-rational $c$ provided the elements of the MTW tensor are rational (see **Examples 2 and 4** in Sec. 4),

- can be used to solve the inverse problem, namely to compute a semialgebraic inner approximation of the region where MTW non-negativity holds (see Sec. 4.2).

Our results open the door for computational verification of OT regularity for a general class of non-Euclidean ground cost $c$.

**Organization.** The remaining of this paper is structured as follows. Sec. 2 provides the background on OT regularity and SOS programming. Sec. 3 details the SOS formulation for the *forward problem*, i.e., certification/falsification of the non-negativity of the MTW tensor (Sec. 3.1), and for the *inverse problem*, i.e., semialgebraic inner approximation of the region where MTW non-negativity holds. In Sec. 4, we illustrate the proposed SOS computational framework for both the forward (**Examples 1 and 2**) and the inverse (**Examples 3 and 4**) problems. Sec. 5 concludes the paper.

## 2 NOTATIONS AND BACKGROUND

**Notations.** A summary of notations is listed in Table 1. Except in the case of ground cost functions $c(x, y)$, we use the subscripts to denote components. For example, $x_i$ denotes the $i$th component of the vector $x$, and $[F]_{i,j}$ denotes the $(i, j)$th component of the matrix $F$.

In the case of cost functions, subscripts such as $c_{ij,kl}(x, y)$ denote the corresponding partial derivative: $\frac{\partial^4}{\partial x_i \partial x_j \partial y_k \partial y_l} c(x, y)$. In this case, the comma in the subscript $ij, kl$, is used to separate indices corresponding to components of variable $x$ and components of variable $y$. The superscripts such as $c^{i,j}(x, y)$ stand for $(i, j)$th element of the inverse of the mixed-Hessian of $c$, i.e., $c^{i,j}(x, y) = [H(x, y)]_{i,j}$ where

$$H := ((\nabla_x \otimes \nabla_y)c)^{-1}. \tag{1}$$

A basic assumption in the regularity theory of OT is that the matrix $(\nabla_x \otimes \nabla_y)c$ is non-singular (Villani, 2009), and thus its inverse, i.e., the matrix $H$, is well-defined.

Table 1: Symbols and notations used throughout this work

| SYMBOL | DESCRIPTION |
|---|---|
| $[\![n]\!]$ | finite set $\{1, \cdots, n\}$ for $n \in \mathbb{N}$ |
| $x_i$ | the $i$th component of vector $x$ |
| $[A]_{i,j}$ | the $(i, j)$th component of matrix $A$ |
| $x^d$ | monomial vector in components of $x \in \mathbb{R}^n$ of degree $d$, |
| | i.e., $(x_1^{\alpha_1} \cdots x_n^{\alpha_n})$ for all valid permutations of $0 \leq \alpha_i \leq d$ such that $\sum_i \alpha_i = d$ |
| $\mathbb{R}_d[x, y]$ | set of polynomials in $x, y$ with real coefficients of degree $\leq d \in \mathbb{N}$, |
| | i.e., $\sum_i a_i x^{\alpha(i)} y^{\beta(i)}$ where $\alpha(i), \beta(i) \in \mathbb{N}^n$ and $\alpha(i) + \beta(i) \leq d \ \forall i$ |
| $\mathbb{R}_{c,d}[x]$ | set of rational functions with numerator polynomial of degree $c$ and |
| | denominator polynomial of degree $d$, i.e., $\left\{ \frac{f}{g} \mid f \in \mathbb{R}_c[x], g \in \mathbb{R}_d[x], c, d \in \mathbb{N} \right\}$ |
| $\overset{n}{\underset{\text{sos}}{\sum}}[x, y]$ | set of $n \times n$ matrix-valued SOS polynomials in $x, y$; $\underset{\text{sos}}{\sum}[x, y] := \overset{1}{\underset{\text{sos}}{\sum}}[x, y]$ |
| $\mathbb{S}^m \ (\mathbb{S}_+^m)$ | set of symmetric (positive semidefinite) matrices of size $m \times m$ |
| $c_{ij,kl}$ | $\partial_{x_i} \partial_{x_j} \partial_{y_k} \partial_{y_l} c(x, y)$; $c_{ij,k} := \partial_{x_i} \partial_{x_j} \partial_{y_k} c(x, y)$, $c_{j,kl} := \partial_{x_j} \partial_{y_k} \partial_{y_l} c(x, y)$ |
| $T_x \mathcal{M}$ | tangent space of the manifold $\mathcal{M}$ at $x$ |
| $T_x^* \mathcal{M}$ | cotangent space of the manifold $\mathcal{M}$ at $x$ |
| $\text{pminor}(F)$ | the set of principal minors of matrix $F$ |
| $|\cdot|$ | absolute value for scalar argument, the cardinality for set argument |
| $\|\cdot\|$ | Euclidean norm |
| $\nabla_x$ | standard Euclidean gradient w.r.t. vector $x$ |
| $\binom{a}{b}$ | coefficient of $x^b$ term in the binomial expansion of $(1 + x)^a$ where $a \geq b, a, b \in \mathbb{N}$ |
| $I_m$ | Identity matrix of size $m \times m$ |

## 2.1 THE REGULARITY PROBLEM OF OPTIMAL TRANSPORT

Kantorovich (1942) proved the existence of solution for a broad class of OT problems where the optimal transport plan is given by a coupling of the measures $\mu$ and $\nu$. This relaxation is known as the Kantorovich problem of OT. However, the existence theory for the Monge OT problem is considerably more subtle. The conditions for existence and $\mu$-a.e. uniqueness of the Monge OT map $\tau_{\text{opt}}$ were established by Brenier (1991) for $c(x, y) = \|x - y\|_2^2$ in Euclidean space, and by Gangbo & McCann (1996) for more general ground costs $c(x, y)$. Roughly speaking, these results state that if the ground cost $c$ is sufficiently smooth and non-degenerate and the measures $\mu, \nu$ are sufficiently regular (for instance, compactly supported and Lebesgue absolutely continuous), then the solution to the Kantorovich problem will also be a solution to the Monge problem. Furthermore, the measurable mapping $\tau_{\text{opt}}$ is then the $c$-subdifferential of a potential function $\phi$, which is the weak solution of the Jacobian equation

$$\det \left( \nabla^2 \phi - A(x, \nabla \phi) \right) = f(x, \phi, \nabla \phi), \quad x \in \mathcal{X}. \tag{2}$$

Here, $A$ is a matrix-valued function derived from the ground cost $c$, and $f$ is a positive function which depends on $c$, $\mu$, and $\nu$. Since this equation involves the determinant of the Hessian $\nabla^2$, it is a Monge-Ampére-type partial differential equation (Benamou et al., 2014).

Once the existence of the Monge OT has been established, it is natural to study the continuity or smoothness of $\tau_{\text{opt}}$, which is equivalent to understanding the qualitative behavior of solutions to (2).

The notion of weak solutions for Monge-Ampére type equations was established by Aleksandrov (1958). Using this theory, it can be shown that the potential $\phi$ will be differentiable a.e., which implies that the optimal transport is well-defined (see De Philippis & Figalli (2014) for a survey on this topic). However, in order to establish the continuity of the map $\tau_{\text{opt}}$ (e.g., a Lipschitz estimate), it is necessary to find an *a priori* $C^2$ estimate for the potential. This problem is known as the *regularity problem of optimal transport*, and provides the primary motivation for our work.

After Brenier (1991), several works (Caffarelli, 1992; Delanoë, 1991; Urbas, 1997) studied the regularity problem for the squared-distance cost in Euclidean space. They established that the transport is smooth whenever the measures are sufficiently smooth and the support of the target measure is

convex. In addition, Caffarelli showed that the non-convexity of the target space provides a global obstruction[1] to establishing smoothness for the transport. These works relied crucially on the classic result from convex analysis that the sub-differential of a convex function is a convex set (Rockafellar, 1970, p. 215), which prevented their arguments from being extended to other cost functions.

In Ma et al. (2005); Trudinger & Wang (2009), the authors introduced a new condition on the ground cost $c$, which is now known as the MTW condition. They showed that this condition was sufficient to establish the continuity of $\tau_{\text{opt}}$, so long as the measures $\mu, \nu$ were sufficiently regular and the target measure satisfied the appropriate notion of convexity. Later, Loeper (2009) discovered that for sufficiently smooth costs, the MTW condition is equivalent to the sub-differential of a $c$-convex function being connected, which establishes the necessity of this condition in the regularity theory. As a result, we say that costs $c$ which satisfy this condition are *regular*. In order to state this condition precisely, we first introduce a quantity referred to as the *MTW tensor or the MTW curvature*.

**Definition 1** (MTW tensor or curvature). *Let $c : \mathcal{X} \times \mathcal{Y} \to \mathbb{R}$ be a cost function where the open sets $\mathcal{X}, \mathcal{Y} \subseteq \mathcal{M}$, and $\mathcal{M}$ is a smooth, compact, connected $n$ dimensional manifold. For $x \in \mathcal{X}$, $y \in \mathcal{Y}$, $\xi \in T_x \mathcal{X}$, and $\eta \in T_y^* \mathcal{Y}$, the MTW Curvature of $c$ is defined as*

$$\mathfrak{S}_{(x,y)}(\xi, \eta) := \sum_{i,j,k,l,p,q,r,s} (c_{ij,p} c^{p,q} c_{q,rs} - c_{ij,rs}) c^{r,k} c^{s,l} \xi_i \xi_j \eta_k \eta_l. \tag{3}$$

Since the pointwise tangent and cotangent spaces of $\mathcal{X}$ and $\mathcal{Y}$ as in Definition 1 are isomorphic to $\mathbb{R}^n$, we may equivalently write $\mathfrak{S}$ in the quadratic form

$$\mathfrak{S}_{(x,y)}(\xi, \eta) = (\xi \otimes \eta)^\top F(x, y)(\xi \otimes \eta). \tag{4}$$

In (4), the symbol $\otimes$ denotes the Kronecker product, and $F(x, y) \in \mathbb{R}^{n^2 \times n^2}$ has entries

$$[F(x, y)]_{i+n(j-1), k+n(l-1)} = \sum_{p,q,r,s} (c_{ij,p} c^{p,q} c_{q,rs} - c_{ij,rs}) c^{r,k} c^{s,l}. \tag{5}$$

**Definition 2** (MTW(0), MTW($\kappa$)). *Consider the notations as in Definition 1. If $\mathfrak{S}_{(\cdot,\cdot)}(\xi, \eta) \geq 0$ for every pair $(\xi, \eta)$ satisfying $\eta(\xi) = 0$, we say that the cost function $c$ satisfies the* weak MTW *condition, or MTW(0). If the stronger condition $\mathfrak{S} \geq \kappa \|\xi\|^2 \|\eta\|^2$ holds for some $\kappa > 0$, then we say that the ground cost $c$ satisfies the* strong MTW *condition, or MTW($\kappa$).*

Naturally, MTW(0) holds if $F(x, y) \succeq 0$ for all pairs $(x, y)$. Note that these definitions are slightly unusual in that $\eta$ and $\xi$ have different basepoints. This contraction is known as the *pseudo-scalar product* (see Definition 2.1 of Figalli et al. (2012) for its formulation in Riemannian manifolds). In this paper, we implicitly use the Euclidean background to evaluate the pseudo-scalar products and refer the readers to Kim & McCann (2010) for details on its formalization for general costs, as well as an interpretation of the MTW tensor in terms of the curvature of a pseudo-Riemannian geometry.

**Definition 3** (Non-negative cost curvature (NNCC)). *(Figalli et al., 2011) Consider the notations as in Definition 2. If we drop the assumption that $\eta(\xi) = 0$ in Definition 2, then a ground cost $c$ satisfying $\mathfrak{S}_{(\cdot,\cdot)}(\xi, \eta) \geq 0$ for every pair $\xi, \eta$, is said to have* non-negative cost curvature *(NNCC).*

Examples, where the MTW or NNCC conditions hold, can be found in Ma et al. (2005); Lee & McCann (2011); Lee & Li (2012); Figalli et al. (2012); Khan & Zhang (2020). However, both the MTW(0) and MTW($\kappa$) conditions are often difficult to analytically verify for generic ground cost $c$.

Beyond regularity of the OT map, both the MTW condition and NNCC condition provide information about the sub-differential structure of $c$-convex function (Loeper, 2009). In recent work, Jacobs & Léger (2020) developed a method for rapidly solving *unregularized* OT problems using a back-and-forth gradient descent. However, this algorithm requires efficiently computing the $c$-conjugate of a function. For the squared-distance cost, fast computation of Legendre transforms depends crucially on the convexity of the subdifferential of a convex function (Lucet, 1997). For cost functions which satisfy NNCC or MTW, it should be possible to develop rapid algorithms for $c$-conjugation, which remains the primary bottleneck for adapting this algorithm to other cost functions. This can be a potential application for certifying NNCC or MTW condition beyond the regularity of OT map.

---

[1]The precise meaning of global obstruction here is somewhat technical, because the transport can be smooth even in cases where the set $\mathcal{Y}$ is not convex. However, given any non-convex $\mathcal{Y}$, it is possible to find smooth measures $\mu$ and $\nu$ so that the transport has discontinuities. Therefore, non-convexity of $\mathcal{Y}$ prevents one from being able to establish an *a priori* estimate for the transport.

## 2.2 SUM-OF-SQUARES PROGRAMMING

The proposed computational framework for solving the forward and inverse problems related to OT regularity, are built on SOS programming ideas outlined below. We refer the readers to Appendix A for additional details and examples. In Sec. 3, we will apply these ideas to a ground cost $c$ for which $\mathfrak{S}$, as defined in Def. 1, is a rational function in variables $(x, y, \eta, \xi) \in \mathcal{X} \times \mathcal{Y} \times T_x\mathcal{X} \times T_y^*\mathcal{Y}$.

SOS programming is a special class of polynomial optimization problems. Generic polynomial optimization problems take the form

$$\min_{x \in \mathbb{R}^n} \quad f(x) \tag{6}$$

$$\text{subject to} \quad x \in \mathcal{C} := \{x \in \mathbb{R}^n \mid g_i(x) \leq 0 \,\forall\, i \in [\![n_g]\!]\},$$

where $f$ and $g_i$ are given multivariate polynomials in vector variable $x$. Problems of the form (6), in general, are computationally intractable because verifying the non-negativity of a multivariate polynomial is NP-hard (Parrilo, 2003, page 4, Ch. 1). The set $\mathcal{C}$ is semialgebraic, as defined next.

**Definition 4** (Semialgebraic set). *(Bochnak et al., 2013, Ch. 2) A set of the form* $\left\{x \in \mathbb{R}^n \mid g(x) \leq 0, g \in \mathbb{R}_{d_g}[x], d_g \in \mathbb{N}\right\}$ *is called* basic semialgebraic. *A semialgebraic set is defined as the finite union of basic semialgebraic sets.*

**Example.** The set of all $3 \times 3$ correlation matrices a.k.a. the *elliptope*

$$\{(x_1, x_2, x_3) \in \mathbb{R}^3 \,|\, \begin{bmatrix} 1 & x_1 & x_2 \\ x_1 & 1 & x_3 \\ x_2 & x_3 & 1 \end{bmatrix} \in \mathbb{S}_+^3\} = \{(x_1, x_2, x_3) \in [-1, 1]^3 \,|\, 2x_1 x_2 x_3 - x_1^2 - x_2^2 - x_3^2 + 1 \geq 0\}$$

is semialgebraic. The set equality follows from Sylvester's criterion (Meyer, 2000, Sec. 7.6): a symmetric matrix is positive semidefinite if and only if its principal minors are all nonnegative.

Semialgebraic sets are known (Tarski, 1998), (Blekherman et al., 2012, Appendix A.4.4) to be stable under finitely many intersections and unions, complement, topological closure, polynomial mappings and Cartesian product.

Introducing a new variable $\gamma \in \mathbb{R}$, we rewrite (6) as

$$\max_{\gamma \in \mathbb{R}} \quad \gamma \tag{7}$$

$$\text{subject to} \quad f(x) - \gamma \geq 0 \quad \forall x \in \mathcal{C} \text{ semialgebraic.}$$

Under the assumption that the semialgebraic set $\mathcal{C} = \{x \in \mathbb{R}^n \mid g_i(x) \leq 0, i \in [\![n_g]\!]\}$ is also *Archimedean*[2], Putinar's Positivstellansatz (Putinar, 1993) allows expressing the non-negativity of the polynomial $f(x) - \gamma$ over $\mathcal{C}$ with an equivalent (see Appendix A.3) SOS representation:

$$f(x) - \gamma = s_0(x) - \sum_{i \in [\![n_g]\!]} s_i(x) g_i(x), \tag{8}$$

where $s_0, s_1, \ldots, s_{n_g} \in \sum_{\text{sos}}[x]$, the set of multivariate SOS polynomials in vector variable $x \in \mathbb{R}^n$. The degree bounds for $s_0, s_1, \ldots, s_{n_g}$ can be found in Nie & Schweighofer (2007b).

Let $d$ be maximum of the degrees of $f, g_1, \ldots, g_{n_g}$, and let $Z_d(x)$ be a column vector of monomials of the form $(1, x, x^2, \cdots, x^d)$ of length $\zeta := \sum_{r=0}^{d} \binom{n+r-1}{r}$. Since the SOS polynomials $s_0, s_1, \ldots, s_{n_g}$ can be parameterized (Parrilo, 2003) by quadratic forms $s_i(x) = Z_d(x)^\top S_i Z_d(x)$, for $S_i \succeq 0$, problem (7) can be written as a semidefinite program (SDP):

$$\max_{\left(\gamma, S_0, S_1, \ldots, S_{n_g}\right) \in \mathbb{R} \times \underbrace{\mathbb{S}_+^\zeta \times \ldots \times \mathbb{S}_+^\zeta}_{n_g + 1 \text{ times}}} \quad \gamma \tag{9}$$

$$\text{subject to} \quad f(x) - \gamma = Z_d(x)^\top S_0 Z_d(x) - \sum_{i \in [\![n_g]\!]} Z_d(x)^\top S_i Z_d(x) g_i(x),$$

---

[2]slightly stronger than compactness, see Appendix A.3 and Laurent (2009)

where the previous polynomial equality constraint is actually a linear equality constraint in the decision variables. Thanks to this SDP representation, existing software such as SOSTOOLS Papachristodoulou et al. (2013), YALMIP Lofberg (2004) or SOSOPT Seiler (2013) can be used to solve SOS tightening of polynomial optimization problems via interior point methods in polynomial time (Alizadeh et al., 1998).

# 3 PROBLEM FORMULATION

We now apply the SOS programming discussed in Sec. 2.2 to the following two problems.

**Forward problem.** Given $c, \mathcal{X}, \mathcal{Y}$ as per Assumption **A1** in Sec. 1, verify if the ground cost $c : \mathcal{X} \times \mathcal{Y} \to \mathbb{R}_{\geq 0}$ satisfies either MTW(0) or MTW($\kappa$) (Definition 2) or NNCC (Definition 3) condition.

**Inverse problem.** Given $c, \mathcal{X}, \mathcal{Y}$ as per Assumption **A1** in Sec. 1, find semialgebraic $\mathcal{U} \times \mathcal{V} \subseteq \mathcal{X} \times \mathcal{Y}$ such that the ground cost $c : \mathcal{U} \times \mathcal{V} \to \mathbb{R}_{\geq 0}$ satisfies either MTW(0) or MTW($\kappa$) (Definition 2) or NNCC (Definition 3) condition.

The SOS formulations for the aforesaid problems detailed next rely on $F$ in (4)-(5), and hence $\mathfrak{S}$, being rational function in $(x, y) \in \mathcal{X} \times \mathcal{Y}$. For this to hold, the ground cost $c$ being rational as in Assumption **A1** is sufficient (see Proposition 13 in Appendix B) but not necessary. Indeed, **Examples 2 and 4** in Sec. 4 consider non-rational ground cost $c$ for which the proposed SOS method can still be used.

## 3.1 FORWARD PROBLEM

Per Assumption **A1**, let the $\mathcal{X} \times \mathcal{Y}$ be as follows for some $d_m, \ell \in \mathbb{N}$.

$$\mathcal{X} \times \mathcal{Y} = \{(x, y) \in \mathbb{R}^n \times \mathbb{R}^n \mid m_i(x, y) \leq 0, \ m_i(x, y) \in \mathbb{R}_{d_m}[x, y] \ \forall i \in [\![\ell]\!]\} \qquad (10)$$

Then, the optimization formulation of the NNCC condition (Definition 3) on the semialgebraic set $\mathcal{X} \times \mathcal{Y}$, can be written as a feasibility problem:

$$\min \quad 0 \qquad (11)$$
$$\text{subject to} \quad \mathfrak{S}_{(x,y)}(\xi, \eta) \geq 0, \quad \forall (x, y) \in \mathcal{X} \times \mathcal{Y}, \ \xi \in T_x \mathcal{X}, \ \eta \in T_y^* \mathcal{Y}.$$

If (11) has a solution, then the NNCC condition is satisfied for all $(x, y) \in \mathcal{X} \times \mathcal{Y}$.

To verify the MTW($\kappa$) condition (Definition 2) for some $\kappa \geq 0$, we need an additional constraint on the pair $(\xi, \eta)$, namely $\eta(\xi) = 0$. The resulting formulation is

$$\min \quad 0 \qquad (12)$$
$$\text{subject to} \quad \mathfrak{S}_{(x,y)}(\xi, \eta) \geq \kappa \|\xi\|^2 \|\eta\|^2, \quad \forall (x, y) \in \mathcal{X} \times \mathcal{Y}, \xi \in T_x \mathcal{X}, \eta \in T_y^* \mathcal{Y} \text{ such that } \eta(\xi) = 0.$$

Verifying the MTW(0) condition is then the special case ($\kappa = 0$) of (12).

If $F$ in (4)-(5) is a rational function, i.e., $F = \frac{F_N}{F_D} \in \mathbb{R}_{N,D}[x, y]$, then problems (11)-(12) are of the form (7). In particular, the NNCC feasibility problem (11) can be tightened to an SOS program as follows (proof in Appendix C).

**Theorem 5** (NNCC forward problem)**.** *Given the semialgebraic set (10) with a ground cost function $c : \mathcal{X} \times \mathcal{Y} \to \mathbb{R}_{\geq 0}$, let $F$ in (5) be of the form $F = \frac{F_N}{F_D} \in \mathbb{R}_{N,D}[x, y]$, $N, D \in \mathbb{N}$. If there exist $s_0, s_1, \ldots, s_\ell \in \sum_{\text{sos}}^{n^2} [x, y]$ such that*

$$\left(F_N(x, y) + F_N^\top(x, y)\right) - s_0(x, y) F_D(x, y) + \sum_{i \in [\![\ell]\!]} s_i(x, y) m_i(x, y) \in \sum_{\text{sos}}^{n^2} [x, y], \qquad (13)$$

*then $c$ satisfies the NNCC condition on $\mathcal{X} \times \mathcal{Y}$.*

A modified version of Theorem 5 can be used to verify the MTW($\kappa$) condition for $\kappa \geq 0$.

**Theorem 6** (MTW($\kappa$) forward problem). *Given the semialgebraic set (10) with a ground cost function* $c : \mathcal{X} \times \mathcal{Y} \to \mathbb{R}_{\geq 0}$*, let F in (5) be of the form* $F = \frac{F_N}{F_D} \in \mathbb{R}_{N,D}[x,y]$*,* $N, D \in \mathbb{N}$*. If there exist* $s_1, \dots, s_\ell \in \sum_{\text{sos}} [x, y, \xi, \eta]$ *and* $t \in \mathbb{R}_{d_t}[x, y, \xi, \eta]$*,* $d_t \in \mathbb{N}$*, such that*

$$(\xi \otimes \eta)^\top \left( F_N(x,y) + F_N^\top(x,y) \right) (\xi \otimes \eta) - \kappa F_D(x,y) \|\xi\|^2 \|\eta\|^2$$
$$+ \sum_{i \in [\![\ell]\!]} s_i(x, y, \xi, \eta) m_i(x, y) + t(x, y, \xi, \eta) \eta^\top \xi \in \sum_{\text{sos}} [x, y, \xi, \eta], \tag{14}$$

*then* $c$ *satisfies the MTW($\kappa$) condition on* $\mathcal{X} \times \mathcal{Y}$ *for some* $\kappa \geq 0$.

Theorems 5 and 6 allow us to replace the constraints in the feasibility problems (11) and (12) with their SOS counterparts (13) and (14), respectively. As a result, both feasibility problems admit an SDP formulation as in (9). In Sec. 4.1, we will illustrate the solutions for these problems via SOS-TOOLS and YALMIP. In Appendix D, we analyze the runtime complexity of the SDP computation for the feasibility problems (13) and (14).

## 3.2 INVERSE PROBLEM

If the constraints in the forward problems (11) or (12) are infeasible, i.e., the NNCC or the MTW($\kappa$) conditions are *falsified* for some $(x, y) \in \mathcal{X} \times \mathcal{Y}$, $\xi \in T_x \mathcal{X}$, $\eta \in T_y^* \mathcal{Y}$, then one may still be interested to know if such conditions hold *locally*. Indeed, the NNCC or the MTW($\kappa$) conditions are not *globally* satisfied in many OT problems. Nevertheless, if we have smooth measures supported on relatively $c$-convex domains (e.g., small balls) within some set $\mathcal{U} \times \mathcal{V} \subseteq \mathcal{X} \times \mathcal{Y}$ where local regularity holds, then the associated Monge OT map $\tau_{\text{opt}}$ will be continuous. This motivates the inverse problems of finding semialgebraic $\mathcal{U} \times \mathcal{V} \subseteq \mathcal{X} \times \mathcal{Y}$ where NNCC or MTW($\kappa$) holds.

Let $\mathcal{X} \times \mathcal{Y}$ be as in (10), and let $\text{vol}$ denote the volume measure. Then, a natural formulation of the NNCC inverse problem is

$$\underset{\mathcal{U} \times \mathcal{V} \subseteq \mathcal{X} \times \mathcal{Y}}{\arg\max} \quad \text{vol}(\mathcal{U} \times \mathcal{V}) \tag{15}$$
$$\text{subject to} \quad \mathfrak{S}_{(u,v)}(\xi, \eta) \geq 0, \quad \forall (u, v) \in \mathcal{U} \times \mathcal{V}, \, \xi \in T_u \mathcal{U}, \, \eta \in T_v^* \mathcal{V}.$$

Likewise, the MTW($\kappa$), $\kappa \geq 0$, inverse problem is

$$\underset{\mathcal{U} \times \mathcal{V} \subseteq \mathcal{X} \times \mathcal{Y}}{\arg\max} \quad \text{vol}(\mathcal{U} \times \mathcal{V}) \tag{16}$$
$$\text{subject to} \quad \mathfrak{S}_{(x,y)}(\xi, \eta) \geq \kappa \|\xi\|^2 \|\eta\|^2, \, \forall (u, v) \in \mathcal{U} \times \mathcal{V}, \, \xi \in T_u \mathcal{U}, \, \eta \in T_v^* \mathcal{V} \text{ such that } \eta(\xi) = 0.$$

The volume maximization objectives in problems (15)-(16) are motivated by our desire to compute "largest" set under-approximators of $\mathcal{X} \times \mathcal{Y}$ where the desired OT regularity conditions hold locally.

To parameterize the decision variable $\mathcal{U} \times \mathcal{V}$ using polynomials, we define $\mathcal{U} \times \mathcal{V}$ as the zero sublevel set of some measurable $V$, i.e., $\mathcal{U} \times \mathcal{V} = \{(x, y) \in \mathcal{X} \times \mathcal{Y} \mid V(x, y) \leq 0\}$ where $V \in \mathbb{R}_d[x, y]$. If $F$ in (4)-(5) is a rational function, i.e., $F = \frac{F_N}{F_D} \in \mathbb{R}_{N,D}[x, y]$, then problem (19) and its MTW($\kappa$) counterpart can be recast in the form (7) by imposing the constraint $-V(x, y) \leq f(x, y)$ for all principal minors $f$ of $F_N$. Then, the problem (15) becomes

$$\underset{V \in \mathbb{R}_d[x,y]}{\max} \quad \text{vol}\,(\mathcal{U} \times \mathcal{V}) \tag{17}$$
$$\text{subject to} \quad m_i(x, y) \leq V(x, y) \quad \forall \, (x, y) \in \mathcal{X} \times \mathcal{Y}, \, i \in [\![\ell]\!].$$
$$V(x, y) + f(x, y) \geq 0, \quad \forall (x, y) \in \mathcal{X} \times \mathcal{Y}, \, f \in \text{pminor}(F_N).$$

Likewise, the problem (16) can be rewritten as

$$\underset{V \in \mathbb{R}_d[x,y]}{\max} \quad \text{vol}\,(\mathcal{U} \times \mathcal{V}) \tag{18}$$
$$\text{subject to} \quad m_i(x, y) \|\xi\|^2 \|\eta\|^2 \leq V(x, y, \xi, \eta) \quad \forall \, (x, y) \in \mathcal{X} \times \mathcal{Y}, \, i \in [\![\ell]\!],$$
$$V(x, y, \xi, \eta) + (\xi \otimes \eta)^\top F_N(x, y)(\xi \otimes \eta) \geq \kappa F_D(x, y) \|\xi\|^2 \|\eta\|^2,$$
$$\forall (x, y) \in \mathcal{X} \times \mathcal{Y}, \xi \in T_x \mathcal{X}, \eta \in T_y^* \mathcal{Y} \text{ such that } \eta(\xi) = 0.$$

To remove the implicit dependence of the objective $\mathrm{vol}\,(\mathcal{U} \times \mathcal{V})$ on $V$, we note that maximizing $\mathrm{vol}\,(\mathcal{U} \times \mathcal{V})$ is equivalent to minimizing $\mathrm{vol}\,((\mathcal{X} \times \mathcal{Y}) \setminus (\mathcal{U} \times \mathcal{V}))$. Furthermore, utilizing the heuristic used in Theorem 2 of Jones (2024) to solve such sublevel set volume minimization problems, we replace $\mathrm{vol}\,((\mathcal{X} \times \mathcal{Y}) \setminus (\mathcal{U} \times \mathcal{V}))$ with the expression

$$\int_\Lambda |V(x,y) - \max_{i \in [\![\ell]\!]} m_i(x,y)| \mathrm{d}x \mathrm{d}y = \int_\Lambda V(x,y) \mathrm{d}x \mathrm{d}y - \int_\Lambda \max_{i \in [\![\ell]\!]} m_i(x,y) \mathrm{d}x \mathrm{d}y.$$

Since the latter term in the above expression is a constant, minimizing the above is equivalent to minimizing $\int_\Lambda V(x,y) \mathrm{d}x \mathrm{d}y$. Although the best choice for the domain of integration is $\Lambda = \mathcal{X} \times \mathcal{Y}$, that choice may not be compact, and thus the integral may be unbounded. Henceforth, we choose *a priori* $\Lambda$ to be some compact subset of $\mathcal{X} \times \mathcal{Y}$. Thus, the NNCC inverse problem (17) becomes

$$\min_{V \in \mathbb{R}_d[x,y]} \quad \int_\Lambda V(x,y) \mathrm{d}x \mathrm{d}y \tag{19}$$
$$\text{subject to} \quad m_i(x,y) \leq V(x,y) \quad \forall\,(x,y) \in \Lambda,\ i \in [\![\ell]\!],$$
$$V(x,y) + f(x,y) \geq 0 \quad \forall (x,y) \in \Lambda,\ f \in \mathrm{pminor}(F_N).$$

The MTW($\kappa$) inverse problem (18) can be reformulated likewise. Since problem (19) is of the form (7), we perform SOS tightening as in Sec. 3.1 to obtain the following (proof in in Appendix C).

**Theorem 7** (NNCC inverse problem). *Given the semialgebraic set (10) with a ground cost function* $c : \mathcal{X} \times \mathcal{Y} \to \mathbb{R}_{\geq 0}$, *let $F$ in (5) be of the form* $F = \frac{F_N}{F_D} \in \mathbb{R}_{N,D}[x,y]$, $N, D \in \mathbb{N}$. *For some compact set* $\Lambda := \{(x,y) \in \mathcal{X} \times \mathcal{Y} \mid \lambda(x,y) \leq 0, \lambda(x,y) \in \mathbb{R}_{d_\lambda}[x,y], d_\lambda \in \mathbb{N}\}$ *chosen a priori, suppose* $V_\pm : \Lambda \to \mathbb{R}$ *solves the optimization problem*

$$\min_{V \in \mathbb{R}_d[x,y]} \quad \int_\Lambda V(x,y) \mathrm{d}x \mathrm{d}y,$$
$$\text{subject to} \quad V(x,y) - m_i(x,y) + r_i(x,y)\lambda(x,y) \in \sum_{\mathrm{sos}}[x,y], \quad \forall\, i \in [\![\ell]\!],$$
$$V(x,y) \pm F_D(x,y) + s_0(x,y)\lambda(x,y) \in \sum_{\mathrm{sos}}[x,y],$$
$$V(x,y) \pm f_j(x,y) + s_j(x,y)\lambda(x,y) \in \sum_{\mathrm{sos}}[x,y], \quad \forall j \in [\![|\mathrm{pminor}(F_N)|]\!],$$
$$s_0(x,y), s_j(x,y), r_i(x,y) \in \sum_{\mathrm{sos}}[x,y] \quad \forall i \in [\![\ell]\!], j \in [\![|\mathrm{pminor}(F_N)|]\!],$$

*where $f_j$ are principal minors of $F_N$. Then, the ground cost $c$ satisfies the NNCC condition on the set* $\{(x,y) \in \Lambda \mid V_+(x,y) \leq 0\} \cup \{(x,y) \in \Lambda \mid V_-(x,y) \leq 0\}$.

Likewise, the MTW($\kappa$) inverse problem can be recast as an SOS program (Thm. 14 in Appendix C). As in Sec. 3.1, such SOS reformulations can be solved via SOSTOOLS and YALMIP. We will illustrate the same in Sec. 4.2.

## 4 NUMERICAL RESULTS

Now we solve the SOS formulations for the forward (Sec. 3.1) and the inverse (Sec. 3.2) problems for ground costs $c$ found in the literature. A visual comparison of the contours for the costs considered in this Section, can be found in Appendix E. All numerical results reported next were obtained by solving the corresponding SOS programs via SOSTOOLS (Papachristodoulou et al., 2013) and YALMIP (Lofberg, 2004) on a HP Spectre laptop with Intel i7-7500U CPU @2.70GHz (4 CPUs) with 16GB RAM. Our SOS implementations leverage the representation of $F$ in Appendix B, Proposition 13, part (iii). Appendix F lists more examples of non-Euclidean OT costs that appeared in the machine learning literature, and are amenable to the proposed method.

### 4.1 FORWARD PROBLEM

**Example 1 (Perturbed Euclidean cost).** In this example, we consider the perturbed Euclidean cost $c(x,y) = \|x - y\|^2 - \varepsilon\|x - y\|^4,\ x, y \in \mathbb{R}^n, \varepsilon > 0$. Lee & Li (2009) analytically showed that for $\varepsilon$ small enough, the MTW(0) condition is satisfied on the semialgebraic set $\mathcal{X} \times \mathcal{Y} := \{(x,y) \in \mathbb{R}^n \times \mathbb{R}^n : \|x - y\| \leq 0.5\}$.

**Algorithm 1** Bisection method to estimate the largest $\varepsilon$ for which MTW(0) holds

---

Choose $\varepsilon_{\text{tol}}$, $\varepsilon_{\min}$ and $\varepsilon_{\max}$
**while** $|\varepsilon_{\min} - \varepsilon_{\max}| > \varepsilon_{\text{tol}}$
**do**
    $\varepsilon \leftarrow (\varepsilon_{\min} + \varepsilon_{\max})/2$
    **if** $\mathfrak{S}(\varepsilon) \succeq 0$ **then**
        $\varepsilon_{\min} \leftarrow \varepsilon$
    **else**
        $\varepsilon_{\max} \leftarrow \varepsilon$
    **end if**
**end while**
**return** $\varepsilon$

---

We estimate the largest perturbation to the Euclidean metric, i.e., the largest $\varepsilon > 0$ such that the cost function $c$ satisfies the NNCC condition when $n = 1$, and the MTW(0) condition when $n \geq 2$. We do so by invoking Theorem 5 and Theorem 6 respectively, to test the feasibility of the associated SOS programs for varying $\varepsilon > 0$. We then estimate the largest $\varepsilon$, denoted as $\varepsilon_{\max}$, for which the NNCC and MTW(0) conditions hold, via bisection search.

For $n = 1$, there are no pairs $\xi, \eta$ such that $\eta(\xi) = 0$ with $\eta$ and $\xi$ both non-zero. In this case, one can verify analytically that for $\varepsilon \leq 2/3$, the MTW tensor is non-negative, i.e., $\mathfrak{S}_{(\cdot,\cdot)}(\xi, \eta) \geq 0$ for any pair of $\xi, \eta$. Our SOS computation per Theorem 5 followed by bisection estimate matches (Table 2 first column) the analytical prediction $\varepsilon_{\max} = 2/3$.

For testing the MTW(0) condition with $n = 2$, we use the SOS formulation per Theorem 6 and perform the same bisection search for $\varepsilon_{\max}$. To reduce the computational complexity, we leveraged translational invariance of $c$ by fixing $x = 0$, and by parameterizing the orthogonal vector-covector pair $(\xi, \eta)$ as $\xi = [a, 1]^\top$ and $\eta = [-1, a]^\top$.

The last row in Table 2 reports the residuals of the corresponding SOS programs, where the residual equals the largest coefficient in the polynomial $\mathfrak{S}_{(x,y)}(\xi, \eta) - s(x, y, \xi, \eta)^\top s(x, y, \xi, \eta)$ where $s$ is the square root of $\mathfrak{S}$ obtained from the SOS program.

Table 2: Numerical results for **Example 1**

| Dimensions, $n$ | 1 | 2 |
|---|---|---|
| $\varepsilon_{\max}$ | 0.67 | $1.05 \cdot 10^{-2}$ |
| Residual | $1.19 \cdot 10^{-7}$ | $4.18 \cdot 10^{-7}$ |

**Example 2 (Log-partition costs).** We now consider a ground cost of the form $c(x, y) = \Psi(x - y)$ where $\Psi$ is the log-partition function of some exponential family. Pal & Wong (2018; 2020) considered the log-partition function of the multinomial distribution and showed that the solutions to OT with the associated cost (i.e., the free energy) could be used to create pseudo-arbitrages (Fernholz, 1999) in stochastic portfolio theory. Khan & Zhang (2020) showed that this $c$ satisfies the MTW(0) condition and derived a regularity theory for the associated OT. More generally, for $c(x, y) = \Psi(x - y)$, the MTW tensor $\mathfrak{S}$ is proportional to the quantity

$$\mathfrak{A}_x(\xi, \eta) = \sum_{i,j,k,l,p,q,r,s} \left( \Psi_{ijp} \Psi^{pq} \Psi_{qrs} - \Psi_{ijrs} \right) \Psi^{rk} \Psi^{sl} \eta_i \eta_j \xi_k \xi_l, \quad x \in \mathcal{X}, \xi \in T_x\mathcal{X}, \eta \in T_x^*\mathcal{X}, \quad (20)$$

which can be interpreted geometrically in terms of the curvature of an associated Kähler metric (Khan & Zhang, 2020, p. 399). We consider the cost $c(x, y) = \Psi_{\text{IsoMulNor}}(x - y)$, where $\Psi_{\text{IsoMulNor}}(x) := \frac{1}{2} \left( -\log x_1 + \sum_{i=2}^{n} x_i^2/x_1 \right)$ is the log-partition function for isotropic multivariate normal distribution, and $x \in \{x \in \mathbb{R}^n \mid x_1 > 0\}$. The regularity of $c(x, y) = \Psi_{\text{IsoMulNor}}(x - y)$ follows from the non-negativity of $\mathfrak{A}$ (Khan & Zhang, 2022, Prop. 9) but checking the latter is non-trivial for $n > 2$ and this computation provides a certificate of this property.

Although $c(x, y) = \Psi_{\text{IsoMulNor}}(x - y)$ is not a rational function, but the inverse of the mixed Hessian $H$ in (1) is a matrix-valued polynomial, and consequently $\mathfrak{A}_x(\xi, \eta)$ is a rational function. Specifically, if we parameterize the vector-covector pairs $(\xi, \eta)$ as $\xi = [\xi_1, \cdots, \xi_n]^\top$, $\eta = [\eta_1, \cdots, \eta_{n-1}, -\frac{1}{\xi_n} \sum_{i=1}^{n-1} \xi_i \eta_i]^\top$, then $\mathfrak{A}_x(\xi, \eta) = \text{poly}(x, \xi, \eta)/x_1^2 \xi_n^2$, where poly denotes a polynomial in $x, \xi, \eta$. For $n = 2$, direct computation gives $\text{poly}(x, \xi, \eta) = 6\xi_1^2 (\xi_2 x_1 - \xi_1 x_2)^2$ which is trivially in SOS form. However, for $n \geq 3$, analytic verification of non-negativity of poly is significantly challenging. In such cases, we use Theorem 6 to find an SOS decomposition of the form $\text{poly}(x, \xi, \eta) = s(x, \xi, \eta)^\top s(x, \xi, \eta)$ via the YALMIP toolbox (Lofberg, 2004). The explicit expression of the polynomial $s$ thus computed for $n = 3$, is reported in Appendix E. In Table 3, we report the residuals and total computational time taken to set up and solve the SOS optimization problem in YALMIP (Lofberg, 2004).

### 4.2 INVERSE PROBLEM

**Example 3 (Perturbed Euclidean cost revisited).**

Table 3: Numerical results for **Example 2**

| Dimensions, $n$ | 3 | 4 | 5 | 6 |
|---|---|---|---|---|
| Residual | $1.034 \cdot 10^{-7}$ | $4.804 \cdot 10^{-8}$ | $4.683 \cdot 10^{-8}$ | $3.475 \cdot 10^{-11}$ |
| Total time (sec) | 0.7220 | 0.8050 | 1.2520 | 1.6690 |

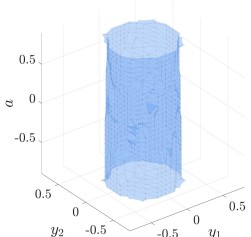

Figure 1: Inner approximation of the region where MTW tensor is $\geq 0$ for **Example 3**.

Recall from Sec. 4.1 that the cost $c(x, y) = \|x - y\|^2 - \varepsilon\|x - y\|^4$ fails the MTW(0) condition for large $\varepsilon > 0$. To illustrate the solution for the MTW(0) inverse problem (Sec. 3.2), we fix $\varepsilon = 1$, $\Lambda = [-1, 1]^2$, $\mathcal{X} = \{[0, 0]\}$, and $\mathcal{X} \times \mathcal{Y}$ as in **Example 1**. As before, we parameterize $(\xi, \eta)$ as $\xi = [a, 1]^\top, \eta = [-1, a]^\top$. We solve the SOS formulation of the inverse problem per Theorem 14 (Appendix C) to estimate the region where the MTW(0) condition is locally satisfied. The resulting region is depicted in Fig. 1. In this example, we parameterize $V$ to be a degree 14-polynomial in $(y_1, y_2, a)$. We find that CPU time for solving the underlying SDP is 0.97s (total time from problem setup to plotting is 115s).

**Example 4 (Squared distance cost for a surface of positive curvature).** We now consider the ground cost $c(x, y) = 3(x_1 - y_1)^2(x_2 + y_2) + 4(x_2^3 + y_2^3) - (4x_2y_2 - (x_1 - y_1)^2)^{\frac{3}{2}}$, which is a scaled squared distance induced by the incomplete Riemannian metric $\mathrm{d}s^2 = x_2(\mathrm{d}x_1^2 + \mathrm{d}x_2^2)$ (Bryant, 2018), which has positive Gaussian curvature. On the diagonal $x = y$, the MTW tensor is proportional to the sectional curvature of the metric, so there is a neighborhood in which MTW(0) holds. Therefore, it is of interest to quantify this region.

This metric has two further properties which make analysis of its MTW tensor more tractable. First, it admits a symmetry in the $x_1$ coordinate, which allows us to set $x_1 = 0$. Second, the scalings $(x_1, x_2) \mapsto (ax_1, ax_2)$ are homotheties of the metric. Therefore, we can scale the metric so that $x_2 = 1$, and so to determine regions for which the MTW condition holds, it suffices to assume $\mathcal{X} = [0, 1]$.

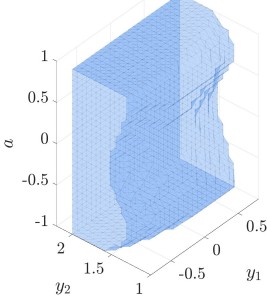

Figure 2: Inner approximation of the region where MTW tensor is $\geq 0$ for **Example 4**.

We fix $\Lambda = [-1, 1] \times [0, 2]$, $\mathcal{X} \times \mathcal{Y} = \{[0, 1]\} \times \{(y_1, y_2) \in [-1, 2] \times [0, 2] \mid 4y_2 - y_1^2 \geq 0\}$, and parameterize the $(\xi, \eta)$ pairs as $\xi = [a, 1]^\top, \eta = [1, -a]^\top$. The MTW tensor $\mathfrak{S}$ is a function of $y_1, y_2, a$. However, since $\mathfrak{S}$ has non-polynomial terms, specifically, $(4y_2 - y_1^2)^{1/2}$, we employ a change of variable $z = (4y_2 - y_1^2)^{1/2}$ to obtain $\mathfrak{S}_{([0,1],y)}(\xi, \eta)$ as a rational polynomial in $y_1, y_2, a, z$. As in **Example 3**, we solve the SOS formulation of the inverse problem per Theorem 14 (Appendix C) to estimate the region where the corresponding MTW(0) condition is locally satisfied. The computed region is shown in Fig. 2. Similar to the previous example, $V$ is parameterized as a degree-14 polynomial in $(y_1, y_2, z, a)$. We find the CPU time for solving the underlying SDP is 19.6s (total time from problem setup to plotting is 120s). The bulk of this time is spent on problem parsing and SDP setup needed to deploy off-the-shelf solvers, and a customized solver should reduce this overhead.

## 5 CONCLUSIONS

We propose a provably correct computational approach to the forward and inverse problems of determining regularity for a cost function based on the sum-of-squares (SOS) programming. The forward problem verifies that a given ground cost globally satisfies non-negative cost curvature (NNCC) or the Ma-Trudinger-Wang (MTW) condition. The inverse problem concerns with finding a region in which the NNCC or the MTW condition holds. The proposed computational approach generalizes for a large class of costs including but not limited to rational functions, and is the first computational work on OT regularity. Our contributions here significantly advance the current state-of-the-art where the NNCC and the MTW conditions have been analytically verified for a limited number of problems. Since the desired conditions require checking the non-negativity of biquadratic forms, analytical approaches have remained unwieldy. We demonstrate that the proposed SOS programming approach can leverage existing solvers, can recover known results in the literature, and can help discover new results on OT regularity.

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

## A   NONNEGATIVE POLYNOMIALS AND SUM-OF-SQUARES PROGRAMMING

In this section, we clarify the connections between the nonnegative and the SOS polynomials. For the benefit of readers not already familiar with this topic, we outline the theoretical rudiments with pedagogical examples.

### A.1   SOS POLYNOMIAL

**Definition 8** (SOS polynomial). *A polynomial* `poly` *in variable* $x \in \mathbb{R}^n$ *is SOS, if there exist finitely many (say $m$) polynomials* $\mathtt{poly}_1, \mathtt{poly}_2, \ldots, \mathtt{poly}_m \in \mathbb{R}[x]$ *such that*

$$\mathtt{poly}(x) = (\mathtt{poly}_1(x))^2 + (\mathtt{poly}_2(x))^2 + \ldots + (\mathtt{poly}_m(x))^2. \tag{21}$$

*We denote the set of all SOS polynomials in* $x \in \mathbb{R}^n$ *as* $\sum_{\mathrm{sos}}[x]$.

A weaker notion of SOS is *nonnegative polynomial*. Specifically, a polynomial `poly` $\in \mathbb{R}[x]$ is called *nonnegative* if $\mathtt{poly}(x) \geq 0$ for all $x \in \mathbb{R}^n$. We note that

$$\sum_{\mathrm{sos}}[x] \subset \text{nonnegative polynomials in } x \subset \mathbb{R}[x]. \tag{22}$$

The following example highlights that the first inclusion is strict.

**Example. (A polynomial that is nonnegative but not SOS)** The Motzkin polynomial (Motzkin, 1967)

$$\mathtt{polyMotzkin}(x_1, x_2) := x_1^4 x_2^2 + x_1^2 x_2^4 - 3x_1^2 x_2^2 + 1,$$

is nonnegative for all $(x_1, x_2) \in \mathbb{R}^2$, as immediate from the AM-GM inequality

$$\frac{x_1^4 x_2^2 + x_1^2 x_2^4 + 1}{3} \geq \left(x_1^4 x_2^2 \cdot x_1^2 x_2^4 \cdot 1\right)^{1/3} = x_1^2 x_2^2.$$

However, the Motzkin polynomial is not an SOS polynomial but instead a ratio of SOS polynomials. The latter follows from that

$$\left(1 + x_1^2 + x_2^2\right) \cdot \mathtt{polyMotzkin}(x_1, x_2)$$

$$= 2\left(\frac{1}{2}x_1^3 x_2 + \frac{1}{2}x_1 x_2^3 - x_1 x_2\right)^2 + \left(x_1^2 x_2 - x_2\right)^2 + \left(x_1 x_2^2 - x_1\right)^2 + \frac{1}{2}\left(x_1^3 x_2 - x_1 x_2\right)^2$$

$$+ \frac{1}{2}\left(x_1 x_2^3 - x_1 x_2\right)^2 + \left(x_1^2 x_2^2 - 1\right)^2.$$

SOS polynomials can be expressed as a quadratic form

$$\mathtt{poly}(x) = (z_d(x))^\top Q z_d(x), \quad Q \succeq 0, \tag{23}$$

where $z_d(x)$ denotes the monomial vector $(1, x, \ldots, x^d)^\top$. In practice, it may suffice to use a subvector of the monomial vector $z_d(x)$.

**Example. (Scalar-valued SOS decomposition)** The SOS polynomial $\mathtt{poly}(x_1, x_2) = 2x_1^4 + 5x_2^4 - x_1^2 x_2^2 + 2x_1^3 x_2$ is expressible as a quadratic form

$$\mathtt{poly}(x_1, x_2) = \begin{pmatrix} x_1^2 \\ x_2^2 \\ x_1 x_2 \end{pmatrix}^\top \begin{bmatrix} 2 & -3 & 1 \\ -3 & 5 & 0 \\ 1 & 0 & 5 \end{bmatrix} \begin{pmatrix} x_1^2 \\ x_2^2 \\ x_1 x_2 \end{pmatrix}.$$

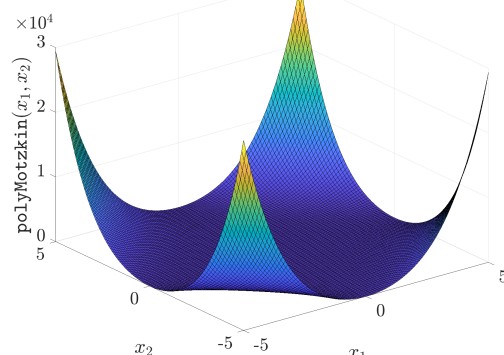

Figure 3: The Motzkin polynomial.

The quadratic form representation called *SOS decomposition*, is convenient because if a nonnegative polynomial is SOS, then certifying its non-negativity is equivalent to certifying that the matrix $Q \succeq 0$. The latter is an SDP feasibility problem amenable to off-the-shelf interior point software.

## A.2 MATRIX-VALUED SOS POLYNOMIAL

A generalization of our interest is $M \times M$ matrix-valued SOS polynomials $\texttt{POLY}(x)$ where $x \in \mathbb{R}^n$. To this end, we first generalize the notion of *nonnegative polynomials* to *semidefinite matrix-valued polynomials* as follows.

**Definition 9** (Semidefinite matrix-valued polynomial). *A mapping*

$$F : \mathbb{R}^n \mapsto \mathbb{S}_+^M$$

*such that all entries of $F(x)$ are in $\mathbb{R}[x]$, is called* positive semidefinite matrix-valued polynomial.

**Definition 10** (Matrix-valued SOS polynomial). *An $M \times M$ positive semidefinite matrix-valued polynomial $\texttt{POLY}$ in variable $x \in \mathbb{R}^n$, is called* matrix-valued SOS polynomial *if it admits a decomposition*

$$\texttt{POLY}(x) = (I_M \otimes z_d(x))^\top Q (I_M \otimes z_d(x)), \quad Q \succeq 0. \tag{24}$$

*We denote the set of all $M \times M$ matrix-valued SOS polynomials in $x \in \mathbb{R}^n$ as $\overset{M}{\underset{\text{sos}}{\sum}}[x]$.*

The matrix-valued SOS decomposition (24) generalizes the scalar-valued SOS decomposition (23).

**Example. (Matrix-valued SOS decomposition)** Consider the matrix-valued polynomial

$$\texttt{POLY}(x_1, x_2) = \begin{bmatrix} 2x_1^2 x_2^2 & x_1 x_2^2 \\ x_1 x_2^2 & 2x_1^2 x_2^2 - 2x_1 x_2^2 + 4x_2^2 \end{bmatrix}.$$

While the elements of $\texttt{POLY}$ are not all SOS polynomials, we can verify that

$$\texttt{POLY}(x_1, x_2) = \left( I_2 \otimes \begin{bmatrix} x_1 x_2 \\ x_2 \end{bmatrix} \right)^\top \begin{bmatrix} 2 & 0 & 0 & 1 \\ 0 & 0 & 0 & 0 \\ 0 & 0 & 2 & -1 \\ 1 & 0 & -1 & 4 \end{bmatrix} \left( I_2 \otimes \begin{bmatrix} x_1 x_2 \\ x_2 \end{bmatrix} \right).$$

Moreover, since the matrix in the above quadratic form is positive semidefinite, $\texttt{POLY}$ is a positive semidefinite matrix-valued polynomial with an SOS decomposition.

In the matrix-valued case, the inclusion (22) generalizes as expected:

$$\overset{M}{\underset{\text{sos}}{\sum}}[x] \subset M \times M \text{ positive semidefinite polynomials in } x \subset \mathbb{R}^{M \times M}[x]. \tag{25}$$

Similar to the case of scalar SOS polynomials, if a matrix-valued semidefinite polynomial is SOS, then certifying so is equivalent to certifying that the matrix $Q$ in (24) satisfies $Q \succeq 0$, which is again an SDP feasibility problem.

## A.3 SOS POLYNOMIALS AND ARCHIMEDEAN SEMIALGEBRAIC SETS

Consider the set of all *nonnegative* polynomials of degree $m$ in $n$ variables. Hilbert (Reznick, 2000) showed that for $n \geq 3, m \geq 6$ or $n \geq 4, m \geq 4$, there exist nonnegative polynomials that are not SOS. No general bound on this gap is known, although there exist related literature for nonnegative polynomials with additional structures (Chesi, 2007; Ahmadi & Parrilo, 2013).

However, the situation is much better in the case of non-negative polynomials over the so-called *Archimedean* semialgebraic sets (Nie & Schweighofer, 2007a; Prestel & Delzell, 2013; Jacobi & Prestel, 2001).

**Definition 11** (Archimedean semialgebraic set). *A compact semialgebraic set $\mathcal{C} := \{x \in \mathbb{R}^n : 0 \leq m_i(x) \in \mathbb{R}[x], i \in [\![\ell]\!]\}$ is called* Archimedean *if there exists $h \in \mathbb{R}[x]$ nonnegative over $\mathcal{C}$, and $s_i \in \sum_{\text{sos}}[x]$ such that*

$$h(x) - \sum_{i=1}^{\ell} s_i(x) m_i(x) \in \sum_{\text{sos}}[x].$$

The reason why the Archimedean property comes in handy in SOS context is as follows. Any polynomial $f$ that is strictly positive on $\mathcal{C} := \{x \in \mathbb{R}^n : 0 \le m_i(x) \in \mathbb{R}[x], i \in [\![\ell]\!]\}$ Archimedean, admits representation

$$f(x) = s_0(x) + \sum_{i=1}^{\ell} s_i(x) m_i(x), \quad \text{where } s_i \in \sum_{\text{sos}}[x].$$

This is equivalent to say that for any $f > 0$ on $\mathcal{C}$ Archimedean, we have

$$f(x) - \sum_i s_i(x) m_i(x) \in \sum_{\text{sos}}[x]$$

since $f(x) - \sum_i s_i(x) m_i(x) = s_0(x) \in \sum_{\text{sos}}[x]$.

We remark here that Definition 11 is non-constructive. In practice, an useful fact is that any compact semialgebraic set $\mathcal{C}$ can be made Archimedean by adding an additional constraint $0 \le m_{\ell+1}(x) = r^2 - x^\top x$ for large enough $r \in \mathbb{R}$. In other words, the positivity of a polynomial on $\mathcal{C}$ semialgebraic, can be verified on $\mathcal{C} \cap \{x : r^2 - x^\top x \ge 0\}$ via SOS programming. In summary, SOS programming formulations are less conservative on compact semialgebraic sets, and not conservative on Archimedean sets. Hence the equivalence between (7) and (8) on Archimedean sets.

## B  SUPPORTING MATHEMATICAL RESULTS

Here, we collect several mathematical results supporting the developments in the main text.

For a suitably smooth ground cost $c$, let the mixed-Hessian $H' := (\nabla_x \otimes \nabla_y)c$. Following (1), we then have

$$H = (H')^{-1} = \begin{bmatrix} c_{1,1} & c_{1,2} & \cdots & c_{1,n} \\ c_{2,1} & c_{2,2} & \cdots & c_{2,n} \\ \vdots & \vdots & \ddots & \vdots \\ c_{n,1} & c_{n,2} & \cdots & c_{n,n} \end{bmatrix}^{-1}, \quad \text{where} \quad c_{i,j} = \frac{\partial^2 c}{\partial x^i \partial y^j}.$$

The Lemma 12 next characterizes $\det(H')$ and the entries of $H$ as rational functions, provided the ground cost $c$ is rational. This will come in handy for establishing Proposition 13 that follows.

**Lemma 12** (Properties of rational cost function). *Let $c \in \mathbb{R}_d[x, y]$ where $x, y \in \mathbb{R}^n$ and $d \in \mathbb{N}$. Let $d_1 := n(d-2)$. Then*

*(i)* $\det(H') \in \mathbb{R}_{d_1}[x, y]$,

*(ii)* $[H]_{i,j} \in \mathbb{R}_{d_1-d+2, d_1}[x, y] \quad \forall(i, j) \in [\![n]\!] \times [\![n]\!]$.

*Similarly, let $c_N/c_D = c \in \mathbb{R}_{N,D}[x, y]$ where $x, y \in \mathbb{R}^n$ and $N, D \in \mathbb{N}$. Let $d_2 := 3D + N - 2$. Then*

*(iii)* $\det(H') \in \mathbb{R}_{nd_2, n4D}[x, y]$,

*(iv)* $[H]_{i,j} \in \mathbb{R}_{4D+(n-1)d_2, nd_2}[x, y] \quad \forall(i, j) \in [\![n]\!] \times [\![n]\!]$.

*Proof.* **Proof of (i).** For any $n \in \mathbb{N}$, we have

$$\det(H') = \sum_{k=1}^{n} c_{k,l} C_{k,l} = \sum_{k=1}^{n} (-1)^{k+l} c_{k,l} M_{k,l}$$

where $C$s and $M$s represent the cofactors and minors of $H'$, respectively. Clearly, $c_{i,j} \in \mathbb{R}_{d-2}[x, y] \; \forall(i, j) \in [\![n]\!] \times [\![n]\!]$.

By induction, we next show that $M_{k,l} \in \mathbb{R}_{d_1-d+2}[x, y]$. Specifically, for $n = 1$, we have $M_{1,1} = 1 \in \mathbb{R}_0[x, y]$. As inductive hypothesis, suppose for $n = N$, we have $M_{k,l} \in \mathbb{R}_{(N-1)(d-2)}[x, y]$ $\forall(k, l) \in [\![N]\!] \times [\![N]\!]$. Now, for the case $n = N + 1$, we get

$$M'_{k,l} = \sum_{k=1}^{N+1} (-1)^{k+l} c_{k,l} M_{k,l} \in \mathbb{R}_{(N+1-1)(d-2)}[x, y],$$

because $c_{k,l} \in \mathbb{R}_{(d-2)}[x,y]$, and the degree of a sum of polynomials is no greater than the maximum degree of the summands. Thus, the inductive hypothesis must hold for any $n \in \mathbb{N}$. Consequently, $\det(H') = \sum_{k=1}^{n}(-1)^{k+l}c_{k,l}M_{k,l} \in \mathbb{R}_{d_1}[x,y]$.

**Proof of (ii).** For $n = 1$,

$$c^{1,1} = [H]_{1,1} = [[c_{1,1}]^{-1}]_{1,1} = 1/c_{1,1} \in \mathbb{R}_{0,(d-2)}[x,y] ,$$

since $c_{1,1} \in \mathbb{R}_{d-2}[x,y]$. Now proceeding by induction, for $n \geq 2$ we have

$$H = \frac{1}{\det(H')}\mathrm{adj}(H') = \frac{1}{\det(H')}\mathrm{cof}(H')^{\top},$$

where $\mathrm{adj},\mathrm{cof}$ denote the adjugate and the cofactor matrix, respectively.

So by part (i) of this Lemma 12, we conclude

$$c^{i,j} = [H]_{i,j} = \frac{C_{j,i}}{\det(H')} = \frac{(-1)^{j+i}M_{j,i}}{\det(H')} \in \mathbb{R}_{(n-1)(d-2),n(d-2)}[x,y],$$

since $M_{j,i}$ is the determinant of an $(n-1) \times (n-1)$ submatrix of $H'$.

**Proof of (iii) and (iv).** We proceed as we did above for the polynomial case. For $n = 1$, $\det(H') = c_{1,1} \in \mathbb{R}_{3D+N-2,4D}[x,y]$, with denominator $c_D^4$. Then the statement (iii) follows by the same inductive argument used for the proof of statement (i).

Similarly, for $n = 1$ we have $c^{1,1} \in \mathbb{R}_{4D,3D+N-2}[x,y]$. Then the statement (iv) follows by proceeding as in the proof for statement (ii), and applying the result (iii), along with cancelling all $c_D^{n-1}$ terms encountered in the numerator and denominator of $c^{i,j}$. $\qquad\square$

We now use Lemma 12 to show that the entries of $F$ in (5) are rational functions (Proposition 13, part (i)-(ii)), under the standing assumption that $c$ is rational. We also derive a generic representation of the entries of $F$ (Proposition 13, part (iii)) that will be helpful for numerical implementation of the proposed SOS formulation.

**Proposition 13** (Entries of $F$). *Consider $F$ as in (5).*

*(i) If $c \in \mathbb{R}_d[x,y]$, $x,y \in \mathbb{R}^n$, $d \in \mathbb{N}$, then*

$$[F]_{i,j} \in \mathbb{R}_{d_N,d_D}[x,y],$$

*where*

$$d_N = 3n(d-2) - d,$$
$$d_D = 3n(d-2).$$

*(ii) If $c_N/c_D = c \in \mathbb{R}_{N,D}[x,y]$, $x,y \in \mathbb{R}^n$, $N,D \in \mathbb{N}$, then*

$$[F]_{i,j} \in \mathbb{R}_{(n^4-1)d_D+d_N,n^4 d_D}[x,y],$$

*where*

$$d_N = 19D + N - 4 + (5N - 2)(3D + N - 2),$$
$$d_D = 12D - 5N(3D + N - 2).$$

*(iii) For given ground cost $c(x,y)$ where $x,y \in \mathbb{R}^n$, define matrices $C \in \mathbb{R}^{n \times n}, D \in (\mathbb{R}^n)^{\otimes 3}$ as $[C]_{r,s} := c_{ij,rs}, [D]_{r,s} := \nabla_x c_{,rs}$. For any $k \in [\![n]\!]$, let $e_k$ denote the kth standard basis vector in $\mathbb{R}^n$. Then*

$$[F]_{i+n(j-1),k+n(l-1)} = (\nabla_y c_{ij})^{\top} H \left((He_k)^{\top}D(He_l)\right) - (He_k)^{\top}C(He_l).$$

*Proof.* **Proof of (i).** By Lemma 12, we have

$$c_{ij,p}, c_{q,rs} \in \mathbb{R}_{d-3}[x,y],$$
$$c_{ij,rs} \in \mathbb{R}_{d-4}[x,y],$$
$$c^{p,q}, c^{r,k}, c^{s,l} \in \mathbb{R}_{(n-1)(d-2),n(d-2)}[x,y].$$

Then, from the arithmetic combinations of rational functions, we find

$$(c_{ij,p}c^{p,q}c_{q,rs} - c_{ij,rs})c^{r,k}c^{s,l} \in \mathbb{R}_{d_N,d_D}[x,y], \tag{26}$$

wherein $d_N, d_D$ are as stated in part (i).

Now observe that regardless of the choice of $p, q, r, s$, the denominator of (26) will be the polynomial $\det(H')$ (as shown in the proof of Lemma 12, part (i)). So, all entries of $F$ are sums of rational polynomials with common denominator and with a numerator degree $d_N$ as above. From this, the desired result follows.

**Proof of (ii).** In the case when $c$ is rational, by Lemma 12, we have

$$c_{ij,p}, c_{q,rs} \in \mathbb{R}_{7D+N-3,8D}[x,y],$$
$$c_{ij,rs} \in \mathbb{R}_{15D+N-4,16D}[x,y],$$
$$c^{p,q}, c^{r,k}, c^{s,l} \in \mathbb{R}_{4D+(n-1)(3D+N-2),n(3D+N-2)}[x,y],$$

where the denominator of all partial derivatives of $c$ is a power of $c_D$. Then, from the arithmetic combinations of the rational terms, cancelling out powers of $c_D$ as allowed, we have

$$(c_{ij,p}c^{p,q}c_{q,rs} - c_{ij,rs})c^{r,k}c^{s,l} \in \mathbb{R}_{d_N,d_D}[x,y],$$

wherein $d_N, d_D$ are as stated. The desired result follows since $[F]_{i,j}$ is a sum of $n^4$ of these rational terms.

**Proof of (iii).** We compute

$$[F]_{i+n(j-1),k+n(l-1)} = \sum_{p,q,r,s}(c_{ij,p}c^{p,q}c_{q,rs} - c_{ij,rs})c^{r,k}c^{s,l}$$

$$= \sum_{r,s}\left((\nabla_y c_{ij})^T H(\nabla_x c_{rs}) - c_{ij,rs}\right)c^{r,k}c^{s,l}$$

$$= \sum_{r,s}(\nabla_y c_{ij})^\top H(\nabla_x c_{rs})c^{r,k}c^{s,l} - \sum_{r,s}c^{r,k}c_{ij,rs}c^{s,l}$$

$$= (\nabla_y c_{ij})^\top H\left(\sum_{r,s}(\nabla_x c_{rs})c^{r,k}c^{s,l}\right) - (He_k)^\top C(He_l)$$

$$= (\nabla_y c_{ij})^\top H\left((He_k)^\top D(He_l)\right) - (He_k)^\top C(He_l).$$

Note that in the last line above, $D$ is considered as a matrix with vectorial elements. $\qquad\square$

## C    PROOFS FOR NNCC AND MTW FORWARD AND INVERSE PROBLEMS

In the following, we provide the proofs of several results which are stated and used in Sec. 3 and 4.

**Proof of Theorem 5.** Suppose such $s_i$ exist. Then, we have that $F_N(x,y) + F_N^T(x,y) \geq 0$ for all $x, y$ such that $F_D(x,y) \geq 0$, $m_i(x,y) \leq 0$. Thus, $F = F_N/F_D$ is non-negative on $\mathcal{X} \times \mathcal{Y}$ and hence, the ground cost $c$ satisfies the NNCC condition. $\qquad\square$

**Proof of Theorem 6.** We proceed similar to the proof of Theorem 5. Suppose such $s_i$ and $t$ exist. Then, $(\xi \otimes \eta)^\top(F_N(x,y) + F_N^\top(x,y))(\xi \otimes \eta) \geq \kappa F_D(x,y)\|\xi\|^2\|\eta\|^2$ for all $(x,y,\xi,\eta)$ such that $m_i(x,y) \leq 0$ and $\eta^\top\xi = 0$. $\qquad\square$

**Proof of Theorem 7.** We present this proof in two parts: one for $V_+$ that solves the SOS problem with a plus sign in the 2nd and 3rd constraints, and the other for $V_-$ with a minus sign therein.

*For the first part*, let $V = V_+$ solve the SOS problem with the plus constraints. Then, for any $x, y$ such that $V(x,y) \leq 0$ and $\lambda(x,y) \leq 0$, we have that $m_i(x,y) \leq V(x,y) \leq 0$ for every $i \in [\![\ell]\!]$ because $r_i(x,y) \geq 0$. Thus, $(x,y) \in \mathcal{X} \times \mathcal{Y}$. On the same zero sublevel set of $V$, we also have that $F_D(x,y) \geq -V(x,y)$ and $f(x,y) \geq -V(x,y) \geq 0$ for all principal minors $f$ of $F_N$

because $s_0(x,y), s_j(x,y) \geq 0$. Thus, $F_D(x,y) \geq 0$, and from Sylvester's criterion, $F_N(x,y) \succeq 0$. Therefore, $F = F_N/F_D \succeq 0$ on the level set $\{(x,y) \in \Lambda \mid V_+(x,y) \leq 0\}$.

*For the second part*, note that $\{(x,y) \in \Lambda \mid V_+(x,y) \leq 0\}$ gives an inner approximation of the region where both $F_N$ and $F_D$ are positive. However, we may have a region where both $F_N$ and $F_D$ are negative. To find the inner approximation of such a region, we use $V_-$ along with the minus constraints in the SOS problem. Thus, the full region where $F_N/F_D$ is non-negative is a union of the two zero sublevel sets $\{(x,y) \in \Lambda \mid V_+(x,y) \leq 0\}$ and $\{(x,y) \in \Lambda \mid V_-(x,y) \leq 0\}$.

Hence, the cost function satisfies NNCC on the set $\{(x,y) \in \Lambda \mid V_+(x,y) \leq 0\} \cup \{(x,y) \in \Lambda \mid V_-(x,y) \leq 0\}$.  $\square$

**Theorem 14** (MTW($\kappa$) inverse problem)**.** *Given the semialgebraic set (10) with a ground cost function $c : \mathcal{X} \times \mathcal{Y} \to \mathbb{R}_{\geq 0}$, let $F$ in (5) be of the form $F = \frac{F_N}{F_D} \in \mathbb{R}_{N,D}[x,y]$, $N, D \in \mathbb{N}$. For some compact set $\Lambda := \{(x,y,\xi,\eta) \mid \lambda(x,y,\xi,\eta) \leq 0, \lambda(x,y,\xi,\eta) \in \mathbb{R}_{d_\lambda}[x,y,\xi,\eta], d_\lambda \in \mathbb{N}\}$ chosen a priori, suppose $V_\pm : \Lambda \to \mathbb{R}$ solves the optimization problem*

$$\min_{V \in \mathbb{R}_d[x,y,\xi,\eta]} \quad \int_\Lambda V(x,y,\xi,\eta)\mathrm{d}x\mathrm{d}y\mathrm{d}\xi\mathrm{d}\eta,$$

$$\text{subject to} \quad V(x,y,\xi,\eta) - m_i(x,y)\|\xi\|^2\|\eta\|^2 + r_i(x,y,\xi,\eta)\lambda(x,y,\xi,\eta) \in \sum_{\text{sos}}[x,y,\xi,\eta],$$

$$V(x,y,\xi,\eta) \pm \left((\xi \otimes \eta)^\top F_N(x,y)(\xi \otimes \eta) - \kappa F_D(x,y)\|\xi\|^2\|\eta\|^2\right)$$
$$+ s_0(x,y,\xi,\eta)\lambda(x,y,\xi,\eta) + t_0(x,y,\xi,\eta)\eta^\top\xi \in \sum_{\text{sos}}[x,y,\xi,\eta],$$

$$r_i(x,y,\xi,\eta), s_0(x,y,\xi,\eta), t_0(x,y,\xi,\eta) \in \sum_{\text{sos}}[x,y,\xi,\eta], \quad \forall\, i \in [\![\ell]\!].$$

*Then, the ground cost $c$ satisfies the MTW($\kappa$) condition on the set $\{(x,y,\xi,\eta) \in \Lambda \mid V_+(x,y,\xi,\eta) \leq 0, \eta(\xi) = 0\} \cup \{(x,y,\xi,\eta) \in \Lambda \mid V_-(x,y,\xi,\eta) \leq 0, \eta(\xi) = 0\}$.*

*Proof.* We proceed similar to the proof of Theorem 7. Suppose $V = V_+$ solves the SOS problem with plus in the 2nd constraint. Then, for any $x, y, \xi, \eta$ such that $V(x,y,\xi,\eta) \leq 0$, $\lambda(x,y,\xi,\eta) \leq 0$ and $\eta(\xi) = 0$, we have that $m_i(x,y) \leq V(x,y,\xi,\eta)/\|\xi\|^2\|\eta\|^2 \leq 0$ for every $i \in [\![\ell]\!]$. Likewise, $\left((\xi \otimes \eta)^\top F_N(x,y)(\xi \otimes \eta) - \kappa F_D(x,y)\|\xi\|^2\|\eta\|^2\right) \geq 0$. Thus, on the zero sublevel set of $V$, we have $(\xi \otimes \eta)^\top F(x,y)(\xi \otimes \eta) \succeq \kappa\|\xi\|^2\|\eta\|^2$ whenever $\eta(\xi) = 0$. Hence, the cost function satisfies MTW($\kappa$) condition on the set $\{(x,y,\xi,\eta) \in \Lambda \mid V_+(x,y,\xi,\eta) \leq 0, \eta(\xi) = 0\}$.

Following similar arguments as in the second part of the proof of Theorem 7, we conclude that $\{(x,y,\xi,\eta) \in \Lambda \mid V_-(x,y,\xi,\eta) \leq 0, \eta(\xi) = 0\}$ gives an approximation of the region satisfying MTW($\kappa$) when $F_N, F_D$ are both negative.

Combining the above, the cost function satisfies the MTW($\kappa$) condition on the union of the sets $\{(x,y,\xi,\eta) \in \Lambda \mid V_+(x,y,\xi,\eta) \leq 0, \eta(\xi) = 0\}$ and $\{(x,y,\xi,\eta) \in \Lambda \mid V_-(x,y,\xi,\eta) \leq 0, \eta(\xi) = 0\}$, as claimed.  $\square$

# D  COMPUTATIONAL COMPLEXITY

Recall that $x^d$ denotes a monomial vector in components of $x \in \mathbb{R}^n$ of degree $d$, and $z_d(x) := (1, x, x^2, \ldots, x^d)^\top$. If a nonnegative (scalar) polynomial `poly` of degree $2d$ in variable $x \in \mathbb{R}^n$, admits an SOS representation

$$\texttt{poly} = (z_d(x))^\top Q z_d(x), \quad Q \succeq 0,$$

then $z_d(x) \in \mathbb{R}^{\binom{d+n}{d}}$; see e.g., Seiler et al. (2013). This can be generalized to matrix-valued SOS polynomials as follows.

Consider an $M \times M$ matrix `POLY` with (not necessarily nonnegative) polynomial entries. If `POLY` admits an SOS representation

$$\texttt{POLY} = (Z_d(x))^\top Q Z_d(x), \quad Z_d(x) := I_M \otimes z_d(x), \quad Q \succeq 0,$$

then the matrix $Q$ has size $M\binom{d+n}{d} \times M\binom{d+n}{d}$. This will be useful in the sequel.

## D.1 COMPLEXITY ANALYSIS FOR THE FORWARD PROBLEM: NNCC

In Theorem 5, to verify the SOS condition (13), we need to find matrices $Q \succeq 0$, $S_0 \succeq 0$, $S_i \succeq 0$ for all $i \in [\![\ell]\!]$, such that

$$
\begin{aligned}
&\left(F_N(x,y) + F_N^\top(x,y)\right) - (I_{n^2} \otimes Z_d(x,y))^\top S_0(I_{n^2} \otimes Z_d(x,y)) F_D(x,y) \\
&+ \sum_{i \in [\![\ell]\!]} (I_{n^2} \otimes Z_d(x,y))^\top S_i(I_{n^2} \otimes Z(x,y)) m_i(x,y) = (I_{n^2} \otimes Z_d(x,y))^\top Q(I_{n^2} \otimes Z_d(x,y)).
\end{aligned}
$$

(27)

This feasibility problem has $(\ell+2)$ positive semidefinite matrices as decision variables. Next, we upper bound the degree of the matrix-valued polynomial in (13), which will allow us to estimate the size of the underlying SDP for complexity analysis. To this end, we will find the number of constraints in the SDP, and the number of decision variables in the same.

**Number of constraints.** Let $m_i \in \mathbb{R}_M[x,y]$ for all $i \in [\![\ell]\!]$. Then, from Proposition 13, we have $[F]_{i,j} \in \mathbb{R}_{(n^4-1)d_D+d_N, n^4 d_D}[x,y]$, where

$$
\begin{aligned}
d_N &= 19D + N - 4 + (5N-2)(3D+N-2) = \mathcal{O}(DN), \\
d_D &= 12D - 5N(3D+N-2) = \mathcal{O}(DN).
\end{aligned}
$$

Notice that $F_N + F_N^\top$ is a $n^2 \times n^2$ matrix-valued polynomial of degree $d_N$. If we parameterize $s_0$ to have of degree $d_N - d_D$, and $s_i$ to be of degree $d_N - M$, then the left-hand-side polynomial in (27) has degree $d_N$ under the practical assumptions that $M \ll \max\{N, D\}$ and $d_D < d_N$. Since the left-hand-side polynomial is in $2n$ variables (because $x, y \in \mathbb{R}^n$) with degree $d_N$, the corresponding monomial vector has length $\binom{d_N+2n}{2n}$. Since two polynomials are equal if and only if their coefficients for all monomials are equal, the number of equality constraints in (27) is $n^4\binom{d_N+2n}{2n}$.

**Number of decision variables.** Since the highest monomial degree in the left-hand-side of (27) is $d_N$ (an even number), the highest degree in $Z_d(x,y)$ must be $d_N/2$. Thus, $Z_d(x,y)$ is a monomial vector of size $\binom{d_N/2+2n}{2n}$. Then the size of all the positive semidefinite matrix decision variables, namely, $Q, S_0, S_1, S_2, \ldots, S_\ell$, is $n^2\binom{d_N/2+2n}{2n} \times n^2\binom{d_N/2+2n}{2n}$. Thanks to symmetry, the number of free variables in each of these matrices, is $n^2\binom{d_N/2+2n}{2n}\left(n^2\binom{d_N/2+2n}{2n}+1\right)/2$. Since the number of the positive semidefinite matrix decision variables is $\ell+2$, the number of decision variables equals

$$
(\ell+2)\, n^2\binom{d_N/2+2n}{2n}\left(n^2\binom{d_N/2+2n}{2n}+1\right)/2. \tag{28}
$$

**Overall complexity.** Given a generic SDP with $M_s$ constraints and an SDP variable of size $N_s \times N_s$, the state-of-the-art worst-case runtime complexity Jiang et al. (2020) is $\mathcal{O}\left(\sqrt{N_s}(M_s N_s^2 + M_s^\omega + N_s^\omega)\right)$ where $\omega \in [2.376, 3]$ is matrix inversion complexity. Notice that this complexity does not account for specific structure of the SDP at hand. For more details, we refer the readers to Nesterov & Nemirovskii (1994); Andersen et al. (2003); Wright (1997).

In our case, (28) equals $N_s(N_s+1)/2$, which gives

$$
N_s = \mathcal{O}\left(\sqrt{\ell}\, n^{2+d_N/2}\right), \tag{29}
$$

where we have suppressed the dependence on constants $N, D$. From the number of constraints discussed before, we have

$$
M_s = \mathcal{O}\left(n^{4+d_N}\right). \tag{30}
$$

Combining (29) and (30) with the aforementioned generic SDP complexity, gives the worst-case runtime complexity

$$
\mathcal{O}\left(\ell^{5/4} n^{9+5d_N/4} + n^{\omega(4+d_N)} + \ell^{\omega/2} n^{\omega(2+d_N/2)}\right). \tag{31}
$$

From (31), we observe that for fixed dimension $n$, the worst-case runtime complexity w.r.t. the number of semialgebraic constraints $\ell$ has faster than linear but sub-quadratic growth. For fixed $\ell$, this runtime complexity has polynomial scaling w.r.t. $n$ with exponent depending on $d_N = \mathcal{O}(DN)$, where $D, N$ denote the degrees of the denominator and numerator polynomials of the cost function $c(x, y)$, respectively. We note that the complexity (31) does not take into account the sparsity pattern induced by the specific block-diagonal structure of the decision variable for our SOS SDP. So the complexity in practice is significantly lower, as observed in the reported numerical examples.

### D.2 Complexity analysis for the forward problem: $\text{MTW}(\kappa)$

In Theorem 6, to verify the SOS condition (14), we need to find $\kappa \geq 0$, $Q \succeq 0$, $S_i \succeq 0$ for all $i \in [\![\ell]\!]$, and suitable real matrix $T$ such that

$$(\xi \otimes \eta)^\top \left( F_N(x, y) + F_N^\top(x, y) \right) (\xi \otimes \eta) - \kappa F_D(x, y) \|\xi\|^2 \|\eta\|^2$$

$$+ \sum_{i \in [\![\ell]\!]} Z_d(x, y, \xi, \eta)^\top S_i Z_d(x, y, \xi, \eta) m_i(x, y) + Z_d(x, y, \xi, \eta)^\top T Z_d(x, y, \xi, \eta) \eta^\top \xi$$

$$= Z_d(x, y, \xi, \eta)^\top Q Z_d(x, y, \xi, \eta). \tag{32}$$

The decision variables for this problem comprises of $(\ell+1)$ positive semidefinite matrices, 1 indefinite matrix, and 1 nonnegative scalar. Similar to the NNCC analysis above, we can estimate the size of SDP problem for the $\text{MTW}(\kappa)$ condition. The key difference here, however, is that the polynomials are now defined on $x, y, \xi, \eta$, and the matrix-valued SOS constraint is replaced by scalar SOS constraints.

**Number of constraints.** For a polynomial in $4n$ variables (because $x, y, \xi, \eta \in \mathbb{R}^n$) with degree $d_N$, we have $\binom{d_N + 4n}{4n}$ monomials and hence the same number of equality constraints.

**Number of decision variables.** Since the highest monomial degree in the left-hand-side of (27) is $d_N$ (even number for the polynomial to be nonnegative), the highest degree in $Z_d(x, y, \xi, \eta)$ must be $d_N/2$. Thus, $Z_d(x, y, \xi, \eta)$ is a monomial vector of length $\binom{d_N/2 + 4n}{4n}$. Then, the size of all the matrices, namely, $T, Q$ and $S_i$, are $\binom{d_N/2+4n}{4n} \times \binom{d_N/2+4n}{4n}$. Since we have $\ell+1$ positive semidefinite matrices, 1 scalar, and 1 indefinite matrix variable, we have $(\ell + 3)n_Q^2 + 1$ variables. Taking the symmetry of the positive semidefinite matrix variables into account, the total number of decision variables equals

$$1 + \binom{d_N/2 + 4n}{4n} \left( (\ell + 3) \binom{d_N/2 + 4n}{4n} + \ell + 1 \right) / 2. \tag{33}$$

**Overall complexity.** Proceeding similar to the NNCC case, if $N_s$ denotes the number of SDP decision variables, then $N_s(N_s + 1)/2$ equals (33). We then obtain the number of constraints $M_s$ and the number of decision variables $N_s$ as

$$M_s = \mathcal{O}\left( n^{d_N} \right), \quad N_s = \mathcal{O}\left( \sqrt{\ell} n^{d_N/2} \right).$$

Therefore, for the SOS computation associated with the $\text{MTW}(\kappa)$ forward problem, the worst-case runtime complexity for off-the-shelf interior point SDP solver is

$$\mathcal{O}\left( \ell^{5/4} n^{9d_N/4} + \ell^{\omega/2+1/4} n^{(\omega/2+1/4)d_N} \right). \tag{34}$$

For fixed dimension $n$, the above complexity is sub-quadratic w.r.t. the number of semialgebraic constraints $\ell$, as was the case in NNCC. For fixed $\ell$, this complexity has polynomial scaling w.r.t. the dimension $n$ where the exponent depends on $d_N = \mathcal{O}(DN)$, as before. Compared to the NNCC case, the scaling of the worst-case complexity for the $\text{MTW}(\kappa)$ case is slightly better w.r.t. dimension $n$, but slightly worse w.r.t. the number of semialgebraic constraints $\ell$. We again point out that since this worst-case analysis does not take into account the sparsity pattern induced by the specific block-diagonal structure of the decision variables for our case, the complexity in practice is lower, as observed in the numerical examples reported herein.

# E AUXILIARY NUMERICAL RESULTS

**Cost contour comparison.** In Sec. 4, **Examples 1 through 4**, we considered non-Euclidean ground costs $c$ to illustrate the utility of the proposed SOS framework in the analysis of OT regularity. To help visualize the deviation of these ground costs from the canonical Euclidean cost, in Fig. 4, we plot the cost to move the unit mass from a fixed (the origin) to an arbitrary location in its neighborhood in $n = 2$ dimensions, and compare it with the squared Euclidean cost.

Figure 4: Comparing contour plots of various ground costs $c(x, 0)$ in Sec. 4 vis-à-vis the squared Euclidean cost in $n = 2$ dimensions. *From left to right*: squared Euclidean cost, perturbed squared Euclidean cost (**Examples 1 and 3**), log-partition cost (**Example 2**), squared distance cost for a surface with positive curvature (**Example 4**).

**SOS decomposition for $\mathfrak{A}_x$ in the case $n = 3$.** In Sec. 4.1, **Example 2**, we illustrated the solution of the MTW(0) forward problem for a log-partition cost given by

$$c(x, y) = \Psi_{\text{IsoMulNor}}(x - y), \quad \text{where } \Psi_{\text{IsoMulNor}}(x) := \frac{1}{2}\left(-\log x_1 + \sum_{i=2}^{n} x_i^2 / x_1\right).$$

We mentioned that for $n \geq 3$, certifying non-negativity for $\mathfrak{A}_x$ is challenging because the resulting expression for $\mathfrak{A}_x$, although rational, is cumbersome to tackle analytically. To support this claim, we report that our computationally discovered SOS decomposition:

$$\text{poly}(x, \xi, \eta) = s(x, \xi, \eta)^\top s(x, \xi, \eta),$$

where $\mathfrak{A}_x(\xi, \eta) = \text{poly}(x, \xi, \eta)/x_1^2\xi_2^2$ in the case $n = 3$, is given by

$$s(x, \xi, \eta) = \begin{bmatrix} 0 & -1.4 & 0 & 0.24 & 0 & 0 \\ 2.4 & 0 & -0.17 & 0 & 0 & 0 \\ 0 & 1.4 & 0 & -0.24 & 0 & 0 \\ -2.4 & 0 & 0.17 & 0 & -0.0002 & 0 \\ 0 & -1.4 & 0 & 0.25 & 0 & -1.2 \\ 2.4 & 0 & -0.17 & 0 & -0.0002 & 0 \\ -1.6 & 0 & -1.9 & 0 & 0 & 0 \\ 0 & 0.52 & 0 & 1.3 & 0 & 0 \\ 0 & 1.4 & 0 & -0.25 & 0 & 0 \\ -0.84 & 0 & 2 & 0 & 0 & 0 \\ 0 & -0.52 & 0 & -1.3 & 0 & 0 \end{bmatrix} \begin{bmatrix} \eta_1\xi_1^2\xi_2 \\ \eta_1\xi_1^2\xi_3 \\ \eta_1\xi_1\xi_2\xi_1 \\ \eta_1\xi_1\xi_3\xi_1 \\ \eta_2\xi_1\xi_2\xi_2 \\ \eta_2\xi_1\xi_2\xi_3 \end{bmatrix}.$$

The above decomposition was obtained by implementing our SOS formulation per Theorem 6 via the YALMIP toolbox (Lofberg, 2004). For brevity, we omitted coefficients of order $10^{-5}$ or lower in the above polynomial.

# F EXAMPLES OF OT WITH NON-EUCLIDEAN GROUND COST

OT is increasingly being used for non-Euclidean manifolds/costs. In Table 4, we report several such instances from the literature. We specifically note that the $F$ corresponding to most of these non-Euclidean costs $c$ indeed have rational entries, thereby amenable to the SOS computation proposed in this paper.

| Manifold | $c(x,y)$ | $F$ | Theory ref. | Application/ML ref. |
|---|---|---|---|---|
| $\mathbb{S}^{n-1} \times \mathbb{S}^{n-1}$ | $-\log\|x-y\|$ | rational | (Wang, 1996; Loeper, 2011) | (Froese Hamfeldt & Turnquist, 2021) |
| $\mathbb{S}^{n-1} \times \mathbb{S}^{n-1}$ | $b_1 - \sqrt{b_2 + b_3\|x-y\|^2}$ | rational under a variable change | (Oliker, 2011) | (Yadav et al., 2019) |
| $\mathbb{R}^n \times \mathbb{S}^{n-1}$ | $-\langle x,y\rangle/\|x\|$ | rational | (Wilson et al., 2014) | (Fan et al., 2021) |
| $\mathbb{H}^n \times \mathbb{H}^n$ | $-\cosh \circ d_{\mathbb{H}^n}(x,y) =$ $-\left(1 + 2\frac{\|x-y\|^2}{(1-\|x\|^2)(1-\|y\|^2)}\right)$ | rational | (Lee & Li, 2012) | (Alvarez-Melis et al., 2020) (Hoyos-Idrobo, 2020) (Vinh Tran et al., 2020) |
| $\mathbb{R}^n \times \mathbb{R}^n$ | $\inf_{\gamma \in \mathcal{C}(x,y)} \int_0^1 \mathcal{L}(\gamma_t, \dot{\gamma}_t)\,\mathrm{d}t$ | - | Lee & Mc-Cann (2011) | Pooladian et al. (2024) |
| Unknown | $\frac{1}{2}d^2(x,y)$ learnt numerically from data | - | - | (Solomon et al., 2015) (Huguet et al., 2023) |
| $\mathbb{R}^n \times \mathbb{R}^n$ | $\log(1+\sum_{i=1}^n \exp(x_i - y_i))$ | rational | Pal & Wong (2018) | Campbell & Wong (2022) |

Table 4: Instances of non-Euclidean OT in the literature. The symbol $d^2(x,y)$ denotes the squared Geodesic distance between $x$ and $y$. In the third row, the constants $b_1, b_2, b_3 > 0$. In the sixth row, $\mathcal{L}$ denotes a suitable Lagrangain, and $\mathcal{C}(x,y)$ is the set of absolutely continuous curves $\gamma_t$, $t \in [0,1]$, such that $\gamma_0 = x$, $\gamma_1 = y$.

In particular, the first and second rows in Table 4 correspond to different non-Eulcidean $c$ over sphere. These problems originated in reflector antenna design, and were one of the driving forces for the mathematical development in OT regularity.

The $c$ in third row corresponds to cosine similarity between an unnormalized and a normalized vector. This OT formulation was used in (Fan et al., 2021, Sec. 6.1) for unpaired text to image generation.

The fourth row corresponds to OT over hyperbolic manifold with the ML relevance being transport between hierarchical word embedding.

The fifth row of Table 4 corresponds to $c$ that are induced by an action integral for a suitable Lagrangian $\mathcal{L}$, i.e., the $c$ is the value of least action. Pooladian et al. (2024) points out that such $c$ are increasingly common in diffusion models. A specific example of such costs is OT over a time-varying linear dynamical system (Chen et al., 2016a) for which $c(x,y)$ is weighted quadratic in $x,y$.

The sixth row in Table 4 points to ML literature where the $c$ is taken as the half of the squared geodesic distance but the geodesic itself is learnt numerically from short-time asymptotic of the heat kernel on that manifold using Varadhan's formula. In such cases, our method can be applied by performing rational approximation of $c$.

The last row is for log-sum-exp cost inspired by information-theoretic considerations. Here too, our SOS framework is applicable. Our **Example 2** considered a generalization of this cost.

