# OpenReview forum: "Sum-of-Squares Programming for Ma-Trudinger-Wang Regularity of Optimal Transport Maps"
_ICLR.cc/2025/Conference — Submitted to ICLR 2025_

### Official Review · Reviewer_o9bV · 2024-10-17

**Soundness:** 3
**Presentation:** 2
**Contribution:** 3
**Rating:** 5
**Confidence:** 4

**Summary:**

This paper presents a computational method for assessing the regularity structure of optimal transport plans. Specifically, the source and target distribution are continuous distributions on a manifold, the cost is some smooth function, and the regularity in question is the regularity of the pushforward map from source to target distributions. In addition, the computational tool computes the region where the MTW holds.

**Strengths:**

1. The paper presents an under-studied aspect of regularity of Monte map. Moreover, the paper contains numerical algorithm and is practical.
2. The inverse problem is well thought through.
3. The numerics seems promising for a relatively difficult problem in OT.

**Weaknesses:**

1. While it is not an issue for the author per se, it is unfortunate that the SOS condition only applies to semialgebraic sets for the manifold.

2. The writing doesn't seem to include sufficient focus on the gap between the non-negativity condition and the SOS counterpart. It doesn't seem to be clear whether SOS is too strong for this case.

3. The presentation is not clear for this paper. For example, the indicting convention for something like $c_{ij, p}$ is quite confusing. The only mention of $c_{ij, kl}$ in the earlier part is too far away, and the readers can not be expected to find where the notation is and also generalize from $c_{ij, kl}$ to $c_{ij, p}$. This is too confusing for this conference.

4. Theorem 5 seems currently wrong: the function $F$ in (5) is matrix-valued. It doesn't seem correct to somehow assess if this matrix-valued rational function belongs to \sum_{SOS}[x, y]. *Unless this issue is either resolved or explained, this reviewer cannot increase the score above the acceptance threshold.*

5. The author doesn't seem to use certain terms in differentiable geometry correctly. For x, y on different points of $\mathcal{M}$, one cannot directly apply an "inner product"/contraction between the tangent plane on $x$ and the contingent plane of $y$. This issue is resolved, however, if $\mathcal{M}$ is a subset of $\R^{n'}$ and the differentiable structure comes from the Euclidean space. The author is advised to change the writing on this and make sure no further major mistake such as this is made.

Minor comments:
- The logic at line 293 is wrong: \eta(\xi) = 0 should belong to after \forall.

**Questions:**

1. The manuscript doesn't contain (A) a discussion on the computational complexity for checking the NNCC condition and the two types of MTW conditions. Also, the discussion on (B) the complexity of the inverse problem is missing. For (A), the author is advised to provide an analysis. For (B), the author is advised to also provide a runtime in the work, i.e. the time it takes to plot the figures.

2. What is the relationship between Monge OT map (which is Borel) and a Brenier map (which is point to point)? Is regularity of Brenier map, supposing it exists, also something that this formulation can answer?

---

> ### Author Response · Authors · 2024-11-21
> **Response to Reviewer o9bV**
>
> **Response to weaknesses:**
> - (**On gap between non-negative and SOS polynomials**) Thanks for pointing this out. *In the revised manuscript, we have clarified the details regarding this gap in the new Appendix A. Specifically, in Appendix A.3, we now explain the SOS formulation over Archimedean semialgebraic sets* which is what causes the equivalence between (7) and (8).
> - (**On index notation**) We agree. *To improve the clarity of index notation in the revised manuscript, we have added all the related notations $c_{ij,kl}$, $c_{ij,k}$ and $c_{j,kl}$ in the same row in Table 1*. We follow this convention for the partial derivatives of $c$ to be consistent with the literature on the MTW tensor, initiated by the original paper [1].
> - (**On Theorem 5**) Thank you for noticing and highlighting this ambiguity. Indeed, as the reviewer pointed out, the quantity in equation (13) is a matrix-valued polynomial. We have revised the manuscript to use a different notation for matrix-valued SOS. Now, we use $ \displaystyle\sum_{\rm SOS}^n$ to represent $n\times n$ matrix-valued SOS constraints. This new notation is added in Table 1. There we also pointed out the notational reduction: $ \displaystyle\sum_{\rm SOS}^1 = \displaystyle\sum_{\rm SOS}$. *To help the readers, the new Appendix A.1 and A.2 in the revised manuscript explain these ideas in a systematic way with accompanying examples*.
> - (**On contraction**) This contraction is known as the *pseudo-scalar product* (see Definition 2.1 of [2] for its formulation for the square-distance cost). For more general costs, this can be formalized by the Kim-McCann pseudo-Riemannian framework of optimal transport (see Definition 2.3 of [3]). To improve the readability of the paper, we avoided stating our results in terms of the pseudo-Riemannian formulation, instead we implicitly used the ambient Euclidean geometry in order to evaluate the pseudo-scalar products. *To address the reviewer's comments, we have explained this in the revised manuscript in the paragraph preceding Definition 3*.
> - (**On minor comment**) We appreciate the reviewer's careful reading. *In the revised manuscript, we have fixed this issue in equation (12)*.
>
> **Response to questions:**
> - (**On computational complexity**) *Following the reviewer's suggestions, in the revised manuscript, we included a computational complexity analysis for the forward NNCC/MTW problems in the new Appendix D*. We derived the scaling w.r.t. the dimension $n$ and the number of semialgebraic constraints $\ell$. In summary, the SOS worst-case complexity is polynomial in $n$ (see Appendix D for details) and sub-quadratic in $\ell$. *As per the reviewer's suggestion, in Sec. 4.2, we also reported the runtimes for the numerical examples 3 and 4 solving the inverse problem*.
> - (**On regularity of Monge and Brenier OT**) Brenier's theorem states that for sufficiently regular measures, the Monge OT map for the Euclidean squared-distance cost exists and admits polar factorization that Brenier studied. So for sufficiently regular measures, these two maps are the same. This map is point to point as well as Borel (i.e., the preimage of open sets are Borel sets) which is necessary for the pushforward of measures to be well-defined. In general, when Brenier's polar factorization holds, then the Monge OT map is the $c$-subdifferential of a $c$-convex potential. The original MTW paper derives a $C^2$ estimate on this potential (which induces a $C^1$ estimate on the transport). This implies that the Jacobian equation is uniformly elliptic, so higher regularity (e.g., differentiability of the transport) follows using standard elliptic bootstrapping. In other words, for smooth costs/measures the optimal OT map and the potential will both either be infinitely differentiable or else there will be a set of measure zero where the potential is non-differentiable and the transport is discontinuous (see Section 4.5 of [4] for the latter case).
>
> [1] X.-N. Ma, N. S. Trudinger, and X.-J. Wang, “Regularity of potential functions of the optimal transportation problem,” Archive for rational mechanics and analysis, vol. 177, pp. 151–183, 2005.
>
> [2] A. Figalli, L. Rifford, and C. Villani, “Nearly round spheres look convex,” American Journal of Mathematics, vol. 134, no. 1, pp.
> 109–139, 2012.
>
> [3] Y.-H. Kim and R. J. McCann, “Continuity, curvature, and the general covariance of optimal transportation,” Journal of the European
> Mathematical Society, vol. 12, no. 4, pp. 1009–1040, 2010.
>
> [4] G. De Philippis and A. Figalli, “The Monge–Amp`ere equation and its link to optimal transportation,” Bulletin of the American
> Mathematical Society, vol. 51, no. 4, pp. 527–580, 2014.

---

> > ### Comment · Reviewer_o9bV · 2024-11-21
> > **Reply to the author**
> >
> > The reviewer thanks the author for the detailed answer. The reviewer is satisfied with the response.
> >
> > The $O(n^{9d_N/4})$ cost seems quite large even in the case where $ n = 2$, and the reviewer is not sure if this scaling is suitable for this particular machine learning conference. The reviewer will keep the score on this ground, but the reviewer would strongly encourage the author to submit to more suitable venues that focus more on the theoretical complexities of machine learning and less on practical solutions to existing problems.

---

> ### Author Response · Authors · 2024-11-23
> **Response to Reviewer o9bV on scaling and scope in ICLR**
>
> We thank the reviewer for their quick feedback and sincere engagement in this discussion.
>
> Although the theoretical worst-case polynomial scaling of the SOS programming methods (not just our algorithm) seem large, SOS programming has been employed consistently and effectively to solve many large-scale problems in control and optimization — a claim supported by our numerical examples (example 2) and other works exploiting sparsity of SOS optimization [2].
>
> It should be noted that SOS is much faster than cylindrical algebraic decomposition [https://en.wikipedia.org/wiki/Cylindrical_algebraic_decomposition] which returns the regions where a real valued polynomial is positive or negative but runs in doubly exponential time.
>
> Please also note that the complexity analyses in our Appendix D are valid for off-the-shelf generic interior point SDP solvers as used in our numerical examples, but do not account for the sparsity patterns induced by the block diagonal structure specific to our formulations. This is why the runtimes observed in our numerical examples are better than the theoretical worst-case derived in the newly included Appendix D. In practice, additional speed-ups are possible for specific problems by taking into account suitable symmetries of the cost and/or the manifold (e.g., translation and/or rotational invariance), pre-solving the underlying SDP problems to eliminate unnecessary decision variables and constraints, employing relaxations such as diagonally dominant SOS (DSOS) [1] and sparse SOS (SSOS) [2].
>
> We also believe that, in addition to the practical performance of our algorithm, the theoretical contributions providing convex optimization algorithm to address an unsolved problem is significant and within the scope of ICLR. There are no other computational approaches available in the literature for this problem despite its relevance in ML.
>
> Reference:
>
> [1] A. A. Ahmadi and A. Majumdar, “Dsos and sdsos optimization: more tractable alternatives to sum of squares and semidefinite optimization,” SIAM Journal on Applied Algebra and Geometry, vol. 3, no. 2, pp. 193–230, 2019.
>
> [2] Y. Zheng, G. Fantuzzi, and A. Papachristodoulou, “Sparse sum-of-squares (sos) optimization: A bridge between dsos/sdsos and sos optimization for sparse polynomials,” in 2019 American Control Conference (ACC). IEEE, 2019, pp. 5513–5518.

---

> > ### Comment · Reviewer_o9bV · 2024-11-24
> >
> > The reviewer would have to apologize to the author that the reviewer has to maintain the score. The current manuscript is more suitable for more theoretical venues, for example in applied probability. The crux of the matter is the assumptions given by the work, combined with the computational scaling, are too restrictive for practical usage.
> >
> > For OT applications in practice, it is quite well-known that the typical cost function one would pick would be either the squared Euclidean cost $c(x, y) = ||x - y||^2$ or the EMD cost  $c(x, y) = ||x - y||$. For the squared euclidean cost, the regularity is already proven. For the EMD cost, it is quite clear that the assumption A1 doesn't hold. While the authors can perhaps argue that the function admits some rational approximation, the fact remains that it is only just an approximation, and one won't be able to answer regularity questions in a rigorous fashion. If the authors indeed go through this route of a rigorous application of the SOS approach to EMD, then perhaps this work would have been fine in ICLR and the reviewer would support the publication. But the fact of the matter is that this work doesn't address the typical cases in which OT regularity is concerned.
> >
> > In addition, generically, the fundamental theorem of LP says that the optimal transport solution has an almost unique identification, in the sense that a point can at most on average couple with two other points. So what this means is that the regularity and uniqueness in the practical discrete OT case is almost always true. Therefore, the regularity would almost always happen in practice. In practice, non-uniqueness is not a big issue for OT-based ML applications, and the added paragraph in the paper isn't strong enough, as unregularized OT is almost never used due to its daunting computational complexity.
> >
> > Having that as the context, the main contribution of the work is a computational tool to check the regularity that could be efficient in low-dimensions and for quite esoteric rational cost functions. This limitation is quite daunting. If someone presents the cost function of $c(x, y) = || x- y||^2 - ||x-y||^4$, the reaction of a typical researcher would be to just conclude that the lack of convexity means a lack of regularity globally, and a fine-grained approach for figuring out where the regularity fails to hold is very synthetic.
> >
> > Therefore, as much as the reviewer appreciates the work, the argument for the applicability of this work in practice is quite limited. As much as the reviewer appreciates this work, increasing the score beyond the passing threshold would be against the reviewer's professional judgment.

---

> ### Author Response · Authors · 2024-11-25
> **Response to Reviewer o9bV on applicability of the work**
>
> We thank the reviewer for the comments. Some remarks are in order.
>
> **On typical cost functions**
>
> The reviewer is right that historically standard cost functions in OT have been limited to squared Euclidean and EMD. However, the scope of OT is much broader, and in recent years OT in non-Euclidean setting is appearing across ML applications. *In the re-revised manuscript, we have now included Appendix F with a Table listing such costs and references, and pointed out that in most of these settings, our proposed framework applies*.
>
> **On EMD cost**
>
> The reviewer correctly notes that the EMD cost does not satisfy assumption A1, but this particular case is technically subtle. The regularity problem in this setting is still not fully understood: existing partial regularity results are on the ray-monotone OT plan. Many of these proofs in fact consider the cost function $\sqrt{\epsilon+||x-y||^2}$, which approximates the EMD and satisfies the MTW condition. By taking $\epsilon$ to zero, it is possible to recover some weaker regularity results (see, e.g., https://www.sciencedirect.com/science/article/pii/S0021782405000164?via%3Dihub)
>
> Therefore, there are good reasons to consider the MTW tensor even if one only cares about the original Monge cost. This particular approximation can be fit into the SOS framework by introducing auxiliary variables as we did with the **Example 3** in manuscript. Another cost function which approximates the Monge cost is $c(x,y) = ||x-y|| - \epsilon \log(||x-y||)$,  and this satisfies the MTW condition for $||x-y||$ sufficiently large.
>
> >In addition, generically, the fundamental theorem of LP says that the optimal transport solution has an almost unique identification, in the sense that a point can at most on average couple with two other points. So what this means is that the regularity and uniqueness in the practical discrete OT case is almost always true. Therefore, the regularity would almost always happen in practice. In practice, non-uniqueness is not a big issue for OT-based ML applications, and the added paragraph in the paper isn't strong enough, as unregularized OT is almost never used due to its daunting computational complexity.
>
> These statements are referring to the Kantorovich problem of optimal transport, where solutions are unique but split mass. The regularity for practical discrete OT case is almost always *false*, since the OT coupling will not be Monge. Even when one has an efficient way to solve discrete Kantorovich OT, with or without regularization, it is known to be computationally difficult to extract the Monge map from the support of that optimal Kantorovich coupling. The need and difficulties for the same in ML context are noted in refs. cited in lines 35-36 of our Introduction.
>
> **On the importance of solving unregularized OT**
>
> Until very recently, if one wanted to compute discrete OT on a large scale, the only feasible method was to approximate it using a Sinkhorn type algorithm. This development was motivated by the daunting LP complexity which in turn refers to invocation of generic LP solvers. But discrete OT is not a generic LP, and it remains possible to further exploit the structure of the transportation polytope. For instance, the back-and-forth algorithm cited in lines 205-212 is known to solve *unregularized* OT instances faster than the Sinkhorn regularized implementation in POT toolbox (https://pythonot.github.io/). We explained in lines 205-212 why checking NNCC/MTW condition can have broader impact for such algorithm design, beyond checking the regularity of Monge map.

---

### Official Review · Reviewer_fQTp · 2024-10-30

**Soundness:** 3
**Presentation:** 1
**Contribution:** 1
**Rating:** 5
**Confidence:** 2

**Summary:**

In the context of OT, the fourth-order Ma-TrudingerWang (MTW) tensor associated with this ground cost function provides a notion
of curvature. The non-negativity of this tensor plays a crucial role for establishing continuity for the Monge optimal transport map. In general, it is difficult to analytically verify this condition for any given ground cost. This paper proposes a provably correct computational approach which provides certificates of non-negativity for the MTW tensor using Sum-of-Squares (SOS) programming. The authors further show that their SOS technique can also be used to compute an inner approximation of the region where MTW non-negativity holds. They apply this proposed SOS programming method to several practical ground cost functions to approximate the regions of regularity of the corresponding OT maps. They also evaluate the proposed SOS computational framework for both the forward and the inverse problems.

**Strengths:**

This is a mathematically solid paper and resolve an interesting theoretical problem. It proposes a provably correct computational framework that can certify or falsify the non-negativity of the MTW tensor associated with a given ground cost under the assumptions that the ground cost is a rational and semialgebraic function. The proposed approach is based on sum-of-squares (SOS) programming and can be of independent interests. The authors also demonstrate that the proposed computational framework can be applied to non-rational ground cost function given that the elements of the MTW tensor are rational and can be used to solve the inverse problem.

**Weaknesses:**

The main concern is that this paper seems irrelevant to the ICLR community. In practice, the common ground cost function would be Euclidean and I am not really sure if it is practically important to conduct the computational verification of OT regularity for a general class of non-Euclidean ground cost functions. I encourage the authors to elaborate on potential ML applications or benefits of their work on non-Euclidean optimal transport. For example, would you like to discuss how your method could enhance practical ML systems that use OT, or to provide concrete examples of where non-Euclidean costs arise in ML problems?

Another concern is the poor quality of writing. In particular, there are many advanced mathematical notations, such as semialgebraic functions and Archimedean sets, which are not accessible to the ICLR audience. Both Section 2 and Section 3 are written in a technical way without sufficient intuitive explanations or examples alongside the formal mathematical notation. In my humble opinion, the major contribution of the paper would be the SOS formulations for computing the MTW tensors, which is certainly nontrivial, but this paper would much better fit the applied mathematics oriented journal.

**Questions:**

1. I encourage the authors to elaborate on potential ML applications or benefits of their work on non-Euclidean optimal transport. For example, would you like to discuss how your method could enhance practical ML systems that use OT, or to provide concrete examples of where non-Euclidean costs arise in ML problems?

2. I encourage the authors to improve accessibility of technical parts (e.g., the description of forward problems and inverse problems), such as adding more intuitive explanations or examples alongside the formal mathematical notation.

---

> ### Author Response · Authors · 2024-11-21
> **Response to Reviewer fQTp: relevance to ML**
>
> We thank the reviewer for the perceptive comments. Since the weaknesses and questions are on the same two topics, we address them together.
>
> **On relevance to ML, relevance of non-Euclidean ground cost in OT**
>
> - An immediate consequence of having a computational method like ours that can automate certifying OT regularity, is that practitioners will be able to design custom approximation algorithms for the OT map itself with *a priori* quality of approximation guarantees, even with non-Euclidean $c$ on nontrivial geometric domains/manifolds. Currently this is not possible because OT regularity analysis has remained an area where case-by-case hand computation are available for a very limited number of settings, and these results/techniques do not generalize to other settings (costs, manifolds).
>
> - Even when the regularity of the OT map does not hold, still verifying the MTW/NNCC conditions are of importance for designing algorithms for solving the unregulaized OT problems with general costs. *In the revised manuscript, we explain this with relevant citations right before Sec. 2.2*.
>
> - *In the revised manuscript, the new paragraph after assumption **A1** mentions with citation that the historical motivation driving OT regularity theory was in fact the engineering problem of reflector antenna design*, cast as an OT problem with non-Euclidean ground cost $c(x,y)=-\log\|x-y\|$ on the sphere, which is a non-rational cost but for which the entries of the MTW tensor are rational (thus the proposed SOS framework applies).
>
> - OT with non-Euclidean ground costs are common in computer graphics [1] and in high dimensional single cell data analysis [2]. In the computer graphics context, the manifolds are geometric domains/3D surfaces, and the non-Euclidean $c(x,y)$ is the squared geodesic distance over these manifolds. In OT over the single cell data too, the non-Euclidean $c(x,y)$ is induced by the squared geodesic of a (curved) lower dimensional manifold embedded in the (flat) high dimensional state space [3]. Despite being very different applications, both the graphics and the single cell data share the commonality that the squared geodesic, and thus the non-Euclidean $c$ are not analytically available, but are learnt from data via small-time asymptotic of the heat kernel using Varadhan's formula; see [1,2]. The associated Monge-Kantorovich OT problems are then solved with the numerically learnt non-Euclidean $c$, but finding the corresponding OT maps remain challenging [4]. One potential application of our framework can be the following workflow: (i) approximating the non-Eulcidean $c$ via polynomials/rationals, (ii) computationally finding the regularity estimates/domains using the proposed forward/inverse problems, and (iii) then using these estimates to design custom approximants for the corresponding OT maps with guarantees on the quality of approximation.
>
> - Another source of non-Euclidean ground cost is the family of *exponentially concave functions* which has important applications in mathematical finance and information theory [5]. Ref. [6] showed that if one uses the free-energy as a cost function, then the solutions of the associated OT problems will be induced by exponentially concave function. OT regularity theory for this non-Euclidean cost function is available [7] but this serves as a prototypical example of where it is advantageous to generalize the cost function.
>
> [1] J. Solomon, F. De Goes, G. Peyr´e, M. Cuturi, A. Butscher, A. Nguyen, T. Du, and L. Guibas, “Convolutional wasserstein distances:
> Efficient optimal transportation on geometric domains,” ACM Transactions on Graphics (ToG), vol. 34, no. 4, pp. 1–11, 2015.
>
> [2] G. Huguet, A. Tong, M. R. Zapatero, C. J. Tape, G. Wolf, and S. Krishnaswamy, “Geodesic Sinkhorn for fast and accurate optimal
> transport on manifolds,” in 2023 IEEE 33rd International Workshop on Machine Learning for Signal Processing (MLSP). IEEE, 2023,
> pp. 1–6.
>
> [3] G. Huguet, D. S. Magruder, A. Tong, O. Fasina, M. Kuchroo, G. Wolf, and S. Krishnaswamy, “Manifold interpolating optimal-transport
> flows for trajectory inference,” Advances in neural information processing systems, vol. 35, pp. 29 705–29 718, 2022.
>
> [4] G. Schiebinger, J. Shu, M. Tabaka, B. Cleary, V. Subramanian, A. Solomon, J. Gould, S. Liu, S. Lin, P. Berube et al., “Optimal-transport
> analysis of single-cell gene expression identifies developmental trajectories in reprogramming,” Cell, vol. 176, no. 4, pp. 928–943, 2019.
>
> [5] G. Alirezaei and R. Mathar, “On exponentially concave functions and their impact in information theory,” in 2018 Information Theory
> and Applications Workshop (ITA). IEEE, 2018, pp. 1–10.
>
> [6] S. Pal and T.-K. L. Wong, “Exponentially concave functions and a new information geometry,” The Annals of probability, vol. 46,
> no. 2, pp. 1070–1113, 2018.
>
> [7] G. Khan and J. Zhang, “The K¨ahler geometry of certain optimal transport problems,” Pure and Applied Analysis, vol. 2, no. 2, pp.
> 397–426, 2020.

---

> > ### Author Response · Authors · 2024-11-21
> > **Response to Reviewer fQTp: making the writing accessible**
> >
> > **On making the writing accessible**
> >
> > In the revised manuscript, we have made several edits and additions with examples to improve the quality of exposition. These include
> >
> > -  a new Appendix A detailing the ideas related to nonnegative and SOS polynomials, Archimedean sets. This new Appendix A includes several examples to illustrate the progression of ideas, and is referred inline in the main body in Section 2.2,
> >
> > - an example of semialgebraic set right after its definition in the main body in Section 2.2,
> >
> > - a new paragraph after assumption **A1** in the main body in Section 1, better explaining why that assumption is benign.

---

> ### Author Response · Authors · 2024-12-04
> **Closing Response to Reviewer fQTp**
>
> Dear Reviewer fQTp,
>
> If you feel we have adequately addressed the questions and weaknesses in our rebuttal and significantly revised version of the manuscript, kindly consider increasing the score.
>
> *We note in particular the addition of Appendix F in the re-revised manuscript that specifically lists examples of OT with non-Euclidean ground costs, and that they are within the purview of our method -- clarifying a point you raised*.
>
> Thanks again for your time and feedback to help improve our work.
>
> Best regards,
>
> Authors

---

### Official Review · Reviewer_XnX2 · 2024-11-01

**Soundness:** 3
**Presentation:** 3
**Contribution:** 3
**Rating:** 6
**Confidence:** 2

**Summary:**

In this paper, the authors propose the first computational approach to certify the regularity of the Monge map for optimal transport problems with specific conditions on the transport cost and the state spaces. To be more precise, they evaluate the non-negativity of the fourth-order Ma-Trudinger-Wang (MTW) tensor associated to the transport cost, which has been proved to be a sufficient condition to establish the continuity of the Monge transport map under proper conditions on the marginals of the transport plan. In this work, they consider three versions of this non-negativity condition, previously considered by related works. Their method consists in reformulating the MTW condition (for each of the three versions) into a sum-of-squares program defined on a semialgebraic setvia Putinar's Positivstellansatz, which can then be solved with efficient software. In particular, their approach assumes that the transport cost (or at least the corresponding MTW tensor) is a rational function defined over a two-state semialgebraic space (or at least, a two-state space that contains a semialgebraic space). They apply their framework to verify if the transport cost verifies the MTW condition or to find the largest semialgebraic set on which the transport cost verifies the MTW condition. They propose several convincing numeric experiments in small dimension for a large variety of non-trivial transport costs.

**Strengths:**

- Although I am no expert of SOS programming, the paper is well written so that it can be read by a large audience. In particular, the notation is easy to understand and Section 2 provides the most essential theoretical elements from OT and SOS programming domains to introduce the method.
- Given the elements of Section 2, the idea of proving the MTW non negativity via SOS programming seems to be a very good (and natural) idea. This work seems to be the first to answer this question with relatively moderate theoretical and computational frameworks.
- The formulation of the inverse problem is very interesting and once again well introduced and explained.
- The diversity of numerical experiments (i.e. non trivial transport costs) definitely proves the theoretical statements.

**Weaknesses:**

- As it seems crucial to apply SOS programming, the transformation of the non-negativity condition into a SOS representation in Equation (8) would deserve more explanation, in the appendix for example. For non expert readers, this relation is hard to understand.
- The dimension of the numerical experiments is relatively low, while OT aims at solving large-scale problems.
- Although the problem tackled in this paper is interesting from a theoretical perspective, I am quite concerned by the effective application of this work to OT problems.

**Questions:**

- Could you please provide examples of real-world OT applications where the knowledge of the regularity of the Monge map is crucial ?
- Could you please bring more details on the equivalence between non-negativity on polynomial terms and SOS representation ?
- I think it would be of interest to provide the results on the regularity of the Monge transport map given the three types of non-negativity conditions given in the paper.
- Have you considered experiments with higher dimension ? Is there any computational burden ?

---

> ### Author Response · Authors · 2024-11-21
> **Response to Weaknesses: Reviewer XnX2**
>
> We thank the reviewer for all the comments and questions.
>
> **Response to Weaknesses:**
>
> - (**On relation between SOS and non-negativity**) We agree that the Archimedean property and that it enables an equivalence between SOS and polynomial non-negativity is not obvious. *In the revised manuscript, we added an exposition in Appendix A about this circle of ideas. In particular, in the main body of the manuscript, in the sentence containing equation (8), we cited Appendix A.3 which specifically discusses this equivalence*.
>
> - (**On dimension of numerical experiments**) *In the revised manuscript, we have now included Appendix D to explain the computational scaling w.r.t. dimension. This Appendix D details the runtime computational complexity w.r.t. both dimension $n$ and the number of semialgebraic constraints $\ell$*. For the forward problems, the SOS worst-case complexity is polynomial w.r.t. $n$ and sub-quadratic in $\ell$. We point out that our analyses are valid for off-the-shelf generic interior point SDP solvers as used in our numerical examples, but do not account for the sparsity patterns induced by the block diagonal structure specific to our formulations. This is why the runtimes observed in our numerical examples are better than the theoretical worst-case derived in the newly included Appendix D. In practice, additional speed-ups are possible for specific problems by taking into account suitable symmetries of the cost $c(x,y)$ and/or the manifold $\mathcal{M}$ (e.g., translation and/or rotational invariance), pre-solving the underlying SDP problems to eliminate unnecessary decision variables and constraints, employing relaxations such as diagonally dominant SOS (DSOS) [1] and sparse SOS (SSOS) [2].
>
> - (**On application of this work to OT problems**) This work contributes to the OT literature in two ways. First is a direct contribution in the sense automated certification of the MTW/NNCC condition, as demonstrated in our work, will allow the researchers to a priori verify the continuity of the OT map for the specific cost/geometry of interest. This will in turn help design custom approximation algorithms for the OT map with guarantees. Second is an indirect contribution in the sense even when the OT map may fail to be continuous, both the MTW condition and the NNCC condition provide detailed information about the sub-differential structure of $c$-convex function [3]. This can in turn enable the design of fast numerical algorithms [4] for solving *unregularized* OT problems for general cost functions. *In the revised manuscript, we have explained this prospect in the paragraph right before Sec. 2.2*.
>
> [1] A. A. Ahmadi and A. Majumdar, “Dsos and sdsos optimization: more tractable alternatives to sum of squares and semidefinite
> optimization,” SIAM Journal on Applied Algebra and Geometry, vol. 3, no. 2, pp. 193–230, 2019.
>
> [2] Y. Zheng, G. Fantuzzi, and A. Papachristodoulou, “Sparse sum-of-squares (sos) optimization: A bridge between dsos/sdsos and sos
> optimization for sparse polynomials,” in 2019 American Control Conference (ACC). IEEE, 2019, pp. 5513–5518.
>
> [3] G. Loeper, “On the regularity of solutions of optimal transportation problems,” Acta Math, vol. 202, pp. 241–283, 2009.
>
> [4] M. Jacobs and F. L´eger, “A fast approach to optimal transport: The back-and-forth method,” Numerische Mathematik, vol. 146, no. 3,
> pp. 513–544, 2020.

---

> > ### Author Response · Authors · 2024-11-21
> > **Response to Questions: Reviewer XnX2**
> >
> > **Response to Questions:**
> >
> > * (**On OT applications where regularity of Monge map is crucial**) Historically, one of the motivations driving the development of the regularity theory for the Monge map was the engineering problem of reflector antenna design (see [1] cited in manuscript's line 169-170), which is an OT problem with cost $c(x,y) = -\log\|x-y\|$ on the sphere. This is indeed an example where elements of the MTW tensor are rational although $c$ itself is not: a situation where our proposed SOS framework can automatically certify NNCC/MTW conditions. *In the revised manuscript, this motivation is now explicitly stated in the new paragraph added after the assumption **A1***. The regularity knowledge of the OT map is also crucial for designing custom approximation algorithms for the Monge map itself with a priori guarantees on the quality of approximation, and for designing algorithms to solve unregularized OT problems with generic costs [2].
> >
> > * (**On polynomial non-negativity and SOS**) *Per Reviewer's suggestion, we have now included Appendix A in the revised manuscript, which elaborates on the difference between the two problems (namely, optimization problems with polynomial non-negativity constraints and the SOS constraints) and conditions under which the two problems are equivalent.*.
> >
> > * (**On three types of non-negativity conditions**) The precise relationship between the three conditions studied in this paper and the continuity of the optimal transport are the following.
> > 1) The pioneering work [3] used the $MTW(\kappa)$ condition and established that if the measures were sufficiently regular and the supports were relatively $c$-convex, then the OT map would be smooth.
> >  2) In a subsequent paper, [4] showed a similar result for costs which satisfy $MTW(0)$ rather than the previous stronger condition. However, for this result it was necessary to impose stronger assumptions on the supports of the measure (i.e., strong relative $c$-convexity). For this reason, this paper does not cover all of the results proven in the original 2005 paper [3]. Later [1] provided a interpretation of $MTW(0)$ in terms of convex analysis which demonstrates the importance of this condition.
> > 3) NNCC is a strengthening of the MTW condition. Strictly speaking, this condition is not directly related to the regularity of optimal transport except in that it implies the MTW condition. However, it plays an important role in the study of optimal transport because it implies that certain quantities are convex (see Lemma 6.1 of [5]).
> >
> > - (**On computational burden for higher dimensions**)  In example 2, numerical experiments were performed with higher dimensions for the Log-partition cost function. This example is particularly useful in evaluating the runtime complexity of the algorithm because the MTW tensor for this cost is positive definite for all $n$ (but the SOS framework does not know that), and we provide certificates of positivity for all $n$ up to $6$ (findings were summarized in Table 2). In other words, the SOS framework can be used as tool for computational discovery. *In the revised manuscript, we have also provided a thorough runtime complexity analysis in Appendix D*. To summarize, the underlying SDP problem, obtained from the SOS formulation of the forward problem, has polynomial scaling w.r.t. dimension $n$ and sub-quadratic scaling w.r.t. the number of semialgebraic constraints $\ell$.
> >
> > [1] G. Loeper, “On the regularity of solutions of optimal transportation problems,” Acta Math, vol. 202, pp. 241–283, 2009.
> >
> > [2] M. Jacobs and F. L´eger, “A fast approach to optimal transport: The back-and-forth method,” Numerische Mathematik, vol. 146, no. 3, pp. 513–544, 2020.
> >
> > [3] X.-N. Ma, N. S. Trudinger, and X.-J. Wang, “Regularity of potential functions of the optimal transportation problem,” Archive for
> > rational mechanics and analysis, vol. 177, pp. 151–183, 2005.
> >
> > [4] N. S. Trudinger and X.-J. Wang, “On the second boundary value problem for Monge-Amp´ere type equations and optimal transportation,” Annali della Scuola Normale Superiore di Pisa-Classe di Scienze, vol. 8, no. 1, pp. 143–174, 2009.
> >
> > [5] A. Figalli, Y.-H. Kim, and R. J. McCann, “When is multidimensional screening a convex program?” Journal of Economic Theory, vol. 146, no. 2, pp. 454–478, 2011.

---

> > > ### Comment · Reviewer_XnX2 · 2024-11-24
> > > **Answer to the the rebuttal**
> > >
> > > I deeply thank the authors for their significant revision and their clear responses, which helps to improve the readability of the contribution. Due to my unfamiliarity with the domain, I choose to keep my current score.

---

### Official Review · Reviewer_AeCx · 2024-11-04

**Soundness:** 3
**Presentation:** 3
**Contribution:** 3
**Rating:** 6
**Confidence:** 4

**Summary:**

The paper presents a sum-of-squares programming based approach to verifying the Ma-Trudinger-Wang (MTW) condition. Precisely, both the forward problem of identifying if a given cost function and domains satisfy the MTW condition and the inverse problem of finding the largest semialgebraic domain on which the MTW condition holds are considered. The corresponding problems can be solved via standard SOS solvers on modest hardware. The paper concludes with a numerical study which validates the theoretical findings.

**Strengths:**

To my knowledge, this is the first paper which explores the question of numerically verifying the MTW condition. The paper is overall written well, and the theoretical details look correct.

**Weaknesses:**

The limitation of the paper regards the assumption that the cost function is rational or that the elements of the MTW tensor are rational. I believe it would be useful to provide general examples of when it holds/does not hold in the text to further clarify how strong/weak the assumption really is.

I believe, however, that the implications of this work are not quite fully fleshed out.

**Questions:**

My main question pertains to how the authors see this work fitting within the broader optimal transport literature. In effect, though some results require regularity of the optimal transport map to hold, these results typically pertain to questions of statistical estimation. In these settings, population measures are estimated based on samples and so (i) absolute continuity cannot be verified a priori, (ii) upper and lower bounds for the density cannot be verified, and (iii) the support of the distributions are unknown. It is thus unclear to me how the content of the current paper fits within the previous context.

---

> ### Author Response · Authors · 2024-11-21
> **Response to Reviewer AeCx**
>
> We thank the reviewer for the questions and suggestions. Please find our itemized responses below.
>
> **Response to weaknesses:**
>
> The key assumption in our work is that the elements of the MTW tensor are rational, for which the cost function being rational is sufficient, not necessary (stated in the paragraph just before Sec. 3.1). In fact, we provided numerical examples 2 and 4 where $c$ is non-rational, but the elements of the respective MTW tensors are. The cost in example 2 came from stochastic portfolio theory. As another example, the development of OT regularity theory was motivated by the engineering problem of reflector antenna design (see [Loeper, 2009] cited in paragraph before Def. 1), which was cast as an OT problem with cost $c(x,y) = -\log\|x-y\|$. This is indeed an example where elements of the MTW tensor are rational, although $c$ itself is not. Our proposed SOS framework can handle all the aforementioned situations.
>
> Furthermore, many cost functions in practice, e.g., those induced by the squared geodesic on a Riemannian manifold, are either already polynomials/rationals, or smooth enough to be well-approximated by polynomials/rationals. This is why our assumption for the elements of the MTW tensor being rational, is benign.
>
> *Following the reviewer's suggestion, we have added a paragraph clarifying the above the revised manuscript's line 66-72 (highlighted in blue) right after the assumption **A1**.*
>
> **Response to questions:**
>
> It is a fair observation that in many real-world applications, the other hypotheses of the regularity theory will not hold, and thus, the transport may fail to be continuous. However, both the MTW condition and the NNCC condition provide detailed information about the sub-differential structure of $c$-convex function [1]. In particular, a cost function satisfies the MTW condition iff the $c$-subdifferential of any $c$-convex function is connected. A cost function satisfies NNCC iff the $c$-subdifferential of any $c$-convex function is convex in a certain sense. These facts generalize the classic result that the sub-differential of a convex function is a convex set.
>
> In recent work, Jacobs and L\'eger [2] developed a fast method for solving OT problems using a back-and-forth gradient descent. This method is notable because it does not use entropic regularization and converges rapidly. However, this algorithm relies on being able to rapidly compute the $c$-conjugate of a function. For the squared-distance cost, an algorithm for fast computation of Legendre transforms exists but depends crucially on the aforementioned convexity properties [3]. To adapt this algorithm to more general cost functions, the primary bottleneck appears to be a fast method for computing the $c$-subdifferential of a $c$-convex function. For cost functions that satisfy NNCC (or perhaps even MTW), it should be possible to develop rapid algorithms for $c$-conjugation, which would in turn allow us to find efficient algorithms to solve OT problems. This provides an algorithmic motivation behind *a priori* certification of MTW/NNCC, which is what our work is about.
>
> *In the revised manuscript, we have added a paragraph at lines 204 to 212 before Sec. 2.2 explaining this motivation*.
>
> [1] G. Loeper, “On the regularity of solutions of optimal transportation problems,” Acta Math, vol. 202, pp. 241–283, 2009.
>
> [2] M. Jacobs and F. L´eger, “A fast approach to optimal transport: The back-and-forth method,” Numerische Mathematik, vol. 146, no. 3,
> pp. 513–544, 2020.
>
> [3] Y. Lucet, “Faster than the fast Legendre transform, the linear-time Legendre transform,” Numerical Algorithms, vol. 16, pp. 171–185,
> 1997.

---

> > ### Comment · Reviewer_AeCx · 2024-11-22
> >
> > I thank the authors for their response. I believe that this added discussion will be helpful for placing this work in the literature. I will maintain my current score.

---

### Official Review · Reviewer_P2kY · 2024-11-04

**Soundness:** 3
**Presentation:** 3
**Contribution:** 3
**Rating:** 6
**Confidence:** 2

**Summary:**

This paper presents a computational approach using Sum-of-Squares (SOS) programming to verify and approximate regions of regularity for optimal transport maps, specifically focusing on the Ma-Trudinger-Wang (MTW) tensor. Regularity of optimal transport maps is critical in many machine learning applications, and this regularity can be assured by the non-negativity of the MTW tensor. However, verifying this condition analytically for general cost functions is challenging. The authors propose using SOS programming to generate certificates of non-negativity for the MTW tensor across a broader range of cost functions, potentially providing computationally verified regions of regularity. Their method is applied to both verifying non-negativity conditions (the "forward problem") and to computing inner approximations of regularity regions (the "inverse problem") for several ground cost functions, demonstrating the effectiveness of SOS programming in this context. This computational framework contributes a systematic approach to certifying regularity in optimal transport, potentially facilitating its application in various machine learning tasks. The paper concludes by applying the proposed framework to several common examples in the literature.

**Strengths:**

The paper’s strength lies in its innovative application of Sum-of-Squares (SOS) programming to address the longstanding challenge of verifying the regularity of optimal transport (OT) maps through the Ma-Trudinger-Wang (MTW) tensor. SOS programming is a well-established tool in optimization and control, but this work extends it to OT regularity, opening new possibilities for computational verification of the MTW conditions in general cases where analytic approaches are intractable. The paper also showcases the practical efficacy of the approach by applying it to various cost functions, demonstrating its flexibility and adaptability across different scenarios.

**Weaknesses:**

A missing key aspect in the paper is the time complexity analysis for the proposed framework. What's the computational efficiency of SOS programming in verifying regularity of the different OT problems? While the authors showcase the method’s application to specific examples and shared the wall-clock time, a time complexity discussion could be a good addition to the paper.

**Questions:**

1. What are some other regularity verification methods? how does the SOS programming compare to them in terms of accuracy and efficiency?

**Details Of Ethics Concerns:**

No ethics concerns

---

> ### Author Response · Authors · 2024-11-21
> **Response to Reviewer P2kY**
>
> We thank the reviewer for the careful reading and the pertinent comments. Please find our itemized responses below.
>
> **Response to weaknesses:**
>
> Thank you for this suggestion. *In the revised manuscript's Appendix D, we detailed the worst-case runtime complexity analyses for the SDP computation associated with the NNCC forward problem and the MTW$(\kappa)$ forward problem*. We discuss the scaling w.r.t. the dimension $n$ and the number of semialgebraic constraints $\ell$. In summary, the SOS worst-case complexity is polynomial in $n$ and sub-quadratic in $\ell$. We point out that our analyses are valid for off-the-shelf generic interior point SDP solvers as used in our numerical examples but do not account for the sparsity patterns induced by the block diagonal structure specific to our formulations. This is why the runtimes observed in our numerical examples are better than the theoretical worst-case derived in the newly included Appendix D. In practice, additional speed-ups are possible for specific problems by taking into account suitable symmetries of the cost $c(x,y)$ and/or the manifold $\mathcal{M}$ (e.g., translation and/or rotational invariance).
>
> For the inverse problem, in principle, similar analysis is possible. However, the complexity then is governed by the desired tightness of the semialgebraic inner approximation of the region where the NNCC or the MTW$(\kappa)$ conditions hold. *For this reason, in the revised manuscript, we have reported the runtimes for the solution of the inverse problems (numerical examples 3 and 4), as suggested by the Reviewer o9bV*.
>
> **Response to questions:**
>
> This work is the first computational approach on OT regularity verification. All existing results for OT regularity verification in the literature (please see refs. in paragraph before "Contributions" on page 2) are analytical and are available for only a few specific cost functions. Such calculations, when possible, are tedious even for researchers in mathematical analysis and do not generalize for variations in cost functions. This is what motivated our study. As the first computational work in this area, our results show that the proposed SOS framework is extremely promising in terms of accuracy and efficiency in handling non-trivial cost functions, which would be otherwise challenging to do via analytic calculations.

---

> ### Author Response · Authors · 2024-12-04
> **Closing Response to Reviewer P2kY**
>
> Dear Reviewer P2kY,
>
> If you feel we have adequately addressed the questions and weaknesses raised  in our rebuttal and significant revised version of the manuscript, kindly consider increasing the score. Thanks again for your time and feedback to help improve our work.
>
> Best regards,
>
> Authors

---

### Author Response · Authors · 2024-11-21
**Summary of improvements**

We are grateful to the reviewers for their careful reading of our work and for the perceptive inputs provided.

Here is a summary of the major improvements we have made to address the reviewers' comments. For the reviewers' convenience, all revisions in the manuscript are marked in blue.

- **(Accessible writing with pedagogical examples)** Based on the reviewers' comments, we have included a *new Appendix A titled "Nonnegative Polynomials and Sum-of-Squares Programming"* (cited in Sec. 2.2) that explains the SOS polynomial and its decomposition (Sec. A.1), the matrix-valued SOS polynomial and its decomposition (Sec. A.2), and the SOS polynomials and Archimedean semialgebraic sets (Sec. A.3). In addition to definitions and explanations, this includes several examples to illustrate the connections between different ideas (e.g., gap between SOS and nonnegative polynomial). The Sec. A.3 is specifically purposed to explain the equivalence between (7) and (8) for Archimedean semialgebraic sets. In Sec. 2.2, an example is included right after the definition of semialgebraic set. We believe these additions will help the readers who may not be familiar with some of the background ideas.

- **(Computational complexity)** Following the suggestions from multiple reviewers, we performed a detailed computational complexity analysis for the NNCC/MTW forward problems in terms of the dimension $n$ and the number of semialgebraic constraints $\ell$. They scale polynomially in $n$ and sub-quadratic in $\ell$. The analysis is included in *new Appendix D titled "Computational Complexity"* (cited in the end of Sec. 3.1) where Sec. D.1 and D.2 detail the complexity analysis for the NNCC and the MTW forward problems. respectively. In the individual responses to the reviewers, we explain that these results are for off-the-shelf SOS/SDP solvers that do not exploit our problem structure, and thus further speed-ups should be possible by customized solvers. Per suggestion of Reviewer o9bV, the runtimes for the inverse problems are now reported at the end of the numerical examples 3 and 4. Here too, most of the computation overhead was found to be in problem parsing and SDP setup to deploy off-the-shelf solvers. These results are very encouraging for a first computational work on OT regularity.

- **(Improved explanations)** In Sec. 1 (Introduction), we slightly rephrased the 3rd paragraph, included a new paragraph after assumption **A1** to explain why this assumption is benign. The latter also mentions with citation that one of the early driving factors for OT regularity theory was the engineering problem of reflector antenna design which was reformulated as an OT problem with non-Euclidean ground cost $c(x,y)=-\log\|x-y\|$ over the sphere. In Sec. 2.1, paragraph after Definition 2, we clarified some differential geometric issues raised by Reviewer o9bV.

- **(Notational improvement and simplifications)** We fixed the matrix-SOS notation throughout to distinguish it from scalar SOS notation, clarified that as well as the index notation in the revised Table 1. Some additional notations are also included in that Table. Some formulas were simplified in Proposition 13 within Appendix B.

- **(Better positioning the work w.r.t. OT literature)** In addition to mentioning the antenna design problem, we added a new paragraph at the end of Sec. 2.1 explaining why certifying NNCC or MTW condition are of interest in designing algorithms for solving unregularized OT problems with general costs, i.e., such certifications are of interest beyond the regularity of OT map. We wrote a detailed response to Reviewer fQTp on the growing relevance of solving OT problems with non-Euclidean ground costs in ML applications. *We also included the Appendix F listing the OT with non-Euclidean ground cost examples with references, pointing out that most of these are amenable to the proposed SOS framework*.

**Scope and fit for ICLR:**

While SOS programming is well-known in ML and optimization, the novelty this work is its application to automatically certify/falsify OT regularity, and computationally discover regions where local regularity holds. This is an existing problem: its need is recognized in both ML and OT theory literature, but existing approaches rely on unwieldy analytical calculations hand-crafted for very few costs. As a result, these calculations and the related techniques in the existing literature do not generalize. Being the first work on computational verification of OT regularity, this work opens door for a new research direction in computational OT, especially for non-Euclidean OT that is increasingly finding applications in ML including significant ICLR footprints in the recent years. We believe this work--its topic, style, novelty and significance of results--are well within the scope of ICLR.

---

### Meta-Review · Area_Chair_U96f · 2024-12-17

**Metareview:**

The paper presents a computational approach using Sum-of-Squares (SOS) programming to certify the Ma-Trudinger-Wang regularity condition for optimal transport (OT) maps. While technically sound and offering an interesting contribution, the reviewers exposed several weaknesses. A significant concern, particularly emphasized in the final discussions, is the lack of clear exemplification of why this work is relevant for machine learning.. The paper does not convincingly demonstrate how the proposed method provides new insights into ML problems where OT is used or enhances practical ML applications. Moreover, reviewers raised concerns about the high computational complexity of the approach, its limited scalability to higher dimensions, and its focus on semialgebraic cost functions, which may restrict practical applicability. Additional weaknesses include the technical presentation, which lacks accessibility for a broader ML audience, and unclear positioning within the OT and ML literature. Despite the efforts made in revisions, these issues were not sufficiently addressed, and the paper fails to bridge the gap between theoretical contributions and practical relevance.

**Additional Comments On Reviewer Discussion:**

During the rebuttal period, reviewers sought clarifications on the work’s relevance to ML, its scalability, and its computational overhead. While the authors provided detailed responses and additional explanations, the rebuttal did not resolve fundamental concerns about the practical impact of the method and its relevance to the ICLR community. These unresolved issues ultimately led to the decision to reject the paper.

---

### Decision · Program_Chairs · 2025-01-22

Reject